# Lipopolysaccharide -mediated resistance to host antimicrobial peptides and hemocyte-derived reactive-oxygen species are the major *Providencia alcalifaciens* virulence factors in *Drosophila melanogaster*

**Maria Shaka, Aranzazu Arias-Rojas, Alexandra Hrdina, Dagmar Frahm, Igor Iatsenko** *

Max Planck Institute for Infection Biology, Berlin, Germany

* iatsenko@mpiib-berlin.mpg.de

**Data Availability Statement:** All relevant data are within the paper and its Supporting Information files. All raw RNA sequencing data files are

## Abstract

Bacteria from the genus *Providencia* are ubiquitous Gram-negative opportunistic pathogens, causing "travelers' diarrhea", urinary tract, and other nosocomial infections in humans. Some *Providencia* strains have also been isolated as natural pathogens of *Drosophila melanogaster*. Despite clinical relevance and extensive use in *Drosophila* immunity research, little is known about *Providencia* virulence mechanisms and the corresponding insect host defenses. To close this knowledge gap, we investigated the virulence factors of a representative *Providencia* species—*P. alcalifaciens* which is highly virulent to fruit flies and amenable to genetic manipulations. We generated a *P. alcalifaciens* transposon mutant library and performed an unbiased forward genetics screen *in vivo* for attenuated mutants. Our screen uncovered 23 mutants with reduced virulence. The vast majority of them had disrupted genes linked to lipopolysaccharide (LPS) synthesis or modifications. These LPS mutants were sensitive to cationic antimicrobial peptides (AMPs) *in vitro* and their virulence was restored in *Drosophila* mutants lacking most AMPs. Thus, LPS-mediated resistance to host AMPs is one of the virulence strategies of *P. alcalifaciens*. Another subset of *P. alcalifaciens* attenuated mutants exhibited increased susceptibility to reactive oxygen species (ROS) *in vitro* and their virulence was rescued by chemical scavenging of ROS in flies prior to infection. Using genetic analysis, we found that the enzyme Duox specifically in hemocytes is the source of bactericidal ROS targeting *P. alcalifaciens*. Consistently, the virulence of ROS-sensitive *P. alcalifaciens* mutants was rescued in flies with Duox knockdown in hemocytes. Therefore, these genes function as virulence factors by helping bacteria to counteract the ROS immune response. Our reciprocal analysis of host-pathogen interactions between *D. melanogaster* and *P. alcalifaciens* identified that AMPs and hemocyte-derived ROS are the major defense mechanisms against *P. alcalifaciens*, while the ability of the pathogen to resist these host immune responses is its major virulence mechanism. Thus, our work revealed a host-pathogen conflict mediated by ROS and AMPs.

available from the SRA database (accession number PRJNA833403) and can be accessed at https://www.ncbi.nlm.nih.gov/sra/PRJNA833403.

**Funding:** II acknowledges the funding from the Max Planck Society and the Deutsche Forschungsgemeinschaft (DFG) grant IA 81/2-1. The funders had no role in study design, data collection and analysis, decision to publish, or preparation of the manuscript.

**Competing interests:** The authors have declared that no competing interests exist.

## Author summary

Pathogens express special molecules or structures called virulence factors to successfully infect a host. By identifying these factors, we can learn how hosts fight and how pathogens cause infections. Here, we identified virulence factors of the human and fruit fly pathogen *Providencia alcalifaciens*, by infecting flies with a series of mutants of this pathogen. In this way, we detected 23 mutants that were less virulent. Some of these less virulent mutants were hypersensitive to fruit fly immune defense molecules called antimicrobial peptides (AMPs), while others were sensitive to reactive oxygen species (ROS) produced by the immune cells. Notably, AMPs-sensitive mutants remained virulent in a *Drosophila* mutant that lacks AMPs, while pathogens sensitive to oxidative stress retained their virulence in a fruit fly mutant devoid of oxidative species. These results suggest that the ability of *P. alcalifaciens* to resist two major host immune molecules, namely AMPs and ROS, is the major virulence mechanism. Overall, our systematic analysis of *P. alcalifaciens* virulence factors has identified the major defense mechanisms of the fruit fly against this pathogen and the bacterial mechanisms to combat these immune responses.

## Introduction

Host-pathogen conflict is a fundamental biological phenomenon where the success of each interacting organism is dictated by its ability to overcome the other. Consequently, both the host and the microbe [1] contribute to the outcome of any infection, whether it is the death of the host, clearance of the pathogen, or the establishment of a persistent infection. Microbes are equipped with an arsenal of virulence factors that allow them to colonize and survive within the host while causing pathology. In response, hosts have evolved surveillance and defense mechanisms to combat the invading microorganisms. These immune defenses aim not only to clear the pathogen, but also to increase tolerance to a subsequent infection [2,3]. Accordingly, in addition to colonization and invasion factors, microbes must be equipped with strategies that allow them to counteract the host immune responses [4]. Frequently, a subset of the virulence factors expressed by microbes serves this purpose. Exploring the complex interplay between the host immune defenses and counteracting pathogen virulence strategies can help us to comprehend the mechanisms of bacterial pathogenesis and define the role of the host immune defenses in the evolution of pathogen virulence.

One way to explore how the complex interplay between host immune defenses and microbial virulence factors contributes to the outcome of infection is to use *in vivo* model systems amenable to genetic manipulation of both sides of host-pathogen interactions simultaneously [5]. The fruit fly *Drosophila melanogaster* has been extensively used in recent years as a powerful model to study host-microbe interactions. Forward and reverse genetics approaches identified and characterized multiple defense reactions in fruit flies that are conserved throughout evolution [6,7]. The immune system of fruit flies consists of both cellular immune responses, which eliminate foreign invaders via phagocytosis and encapsulation by specialized hemocytes [8], and humoral immune responses leading to the inducible production of antimicrobial peptides (AMPs) and other effector molecules [6,9,10]. Moreover, cuticle injury and recognition of microbial ligands trigger a serine protease cascade resulting in the activation of phenol oxidases and the melanisation reaction. This reaction helps in wound healing and pathogen sequestration via de novo production and deposition of melanin and is assumed to kill pathogens via toxic reactive oxygen species intermediates (ROS) generated during melanisation

[11–13]. Additionally, infection-induced iron sequestration by Transferrin 1 was recently shown to be an essential defense mechanism against certain pathogens [14–16]. Both Tsf1 and the humoral AMP response are regulated by two nuclear factor-kB (NF-κB) pathways: Toll and immune deficiency (Imd) [17]. The Toll pathway is mainly stimulated by Gram-positive bacteria and fungi and is required for the defense against these pathogens [18]. The Toll pathway regulates the expression of antifungal peptides, Drosomycin (Drs) and IM10-like (Baramicins) via the NF-κB Dif and Dorsal [19]. Secreted peptides, the Bomanins, are also under the control of the Toll pathway but their function remains to be understood [20,21]. In contrast, the Imd pathway preferentially responds to Gram-negative bacterial infections, and regulates the expression of different AMPs, like Diptericin (Dpt) or Drosocin (Dro), via the activation of the NF-κB Relish [22].

Previous studies using fruit flies as a model for studying pathogen virulence have shown that the same virulence factors that a pathogen requires to establish infection in a vertebrate host are often also required for pathogenesis in the fly [23–25]. Such observation led to the extensive use of *Drosophila* as a model to study human and animal pathogens [26–29]. The primary aim of such studies was to establish conservation between fly and vertebrate defenses or to dissect the function of known virulence factors. While being fruitful, such vertebrate-centric studies very likely miss new virulence factors, especially those that are insect-specific. The vertebrate-centric use of *Drosophila* as an infection model was also driven by the lack of natural fruit fly pathogens, particularly those that infect flies systemically.

*Providencia* are ubiquitous Gram-negative opportunistic pathogens that have been found in a wide variety of organisms and environments ranging from insects and humans to shark mouths [30–33]. *Providencia rettgeri*, *Providencia alcalifaciens*, and *Providencia stuartii* are considered part of the natural human gut flora but can also be the cause of "travelers' diarrhea" [34–37]. *P. alcalifaciens*, *P. rettgeri*, and *P. stuartii* are the most common clinical isolates, causing urinary tract and other nosocomial infections in humans [30,34,38,39]. Some *P. alcalifaciens* strains, but no currently described strains of the other *Providencia*, can intracellularly invade human cell lines [35,40]. Bacteria of the genus *Providencia* have been isolated as natural infections in wild-caught *D. melanogaster* [41] and in the guts and external environment of *D. melanogaster* [42,43] but they have also been found in various other insects such as medicinal maggots [44], stable flies [45], Mexican fruit flies [46], house flies [47], mediterranean fruit fly [48], *Nasonia* wasps [49], black soldier fly [50], olive fly [51].

The *Providencia* strains that were isolated as pathogens of *D. melanogaster* showed differential potency in virulence [41,52]. *P. sneebia* and *P. alcalifaciens* were highly virulent, killing 90–100% of infected flies, while only 30–40% of infected flies succumbed to the infection with *P. rettgeri* and *P. burhodogranariea* [41]. A comparative genomic analysis of different *Drosophila*-isolated *Providencia* identified isolate-specific genes which might contribute to the variable virulence phenotypes [53]. A similar comparative genomic analysis of clinical *Providencia* isolates revealed a potential genetic basis underlying diversity in pathogenicity, antibiotic resistance and environmental adaptation of *Providencia* [54]. Bacteria from the genus *Providencia* have been widely used to study the mechanisms of *Drosophila* immunity, particularly the specificity of AMPs [55–58], individual variability in infection outcome [59], sexual dimorphism in immunity [60], chronic infections [61], host response to infection [62], and intestinal pathologies during aging [63,64]. Strikingly, despite clinical relevance and extensive use in *Drosophila* immunity research, we know little about *Providencia* virulence mechanisms. To close this knowledge gap, we decided to perform an unbiased genetic screen for *Providencia* virulence factors.

We decided to focus specifically on *P. alcalifaciens* because it is highly virulent in fruit flies and has the best amenability to genetic manipulations among the species that we tested. Given

that *P. alcalifaciens* is both a natural human and fly pathogen, the *Drosophila* model system is particularly useful to study both general *P. alcalifaciens* host-pathogen interactions and insect-specific factors. In this study, we examined the host-pathogen interactions of *P. alcalifaciens* with the insect host, *D. melanogaster*, and identified fly immune mechanisms that are most important for fighting infections with *P. alcalifaciens* as well as the bacterial virulence factors that interact with each of these specific immune responses.

## Results

### *P. alcalifaciens* rapidly kills flies in spite of active immune responses

First, to study *P. alcalifaciens* (*Pa*) virulence we decided to better characterize an infection model and define the experimental conditions, like infectious dose, time points, temperature, etc. We infected $w^{1118}$ DrosDel isogenic ($w^{1118}$ iso) males by needle pricking with different doses and found that *Pa* rapidly kills flies at all tested doses, while there was no mortality from needle injury alone (Fig 1A). Although, there is clear dose-dependent increase in fly mortality, most of the flies did not survive beyond 48h post infection and at OD10 they were all killed within 20h. This is consistent with the previously published high mortality caused by a low dose *Pa* infection [41]. For all further experiments, we chose an infection dose of OD2 (this corresponds to $8.1 \times 10^8$ CFUs/ml in solution and approximately 1500 bacteria per fly) which gives the most optimal resolution in survival between different mutants. Using this dose, we found that different wild-type fly genotypes (*Canton S*, *Oregon R*) succumb quickly to *Pa* infection (Fig 1B), suggesting that high virulence of *Pa* is not restricted to $w^{1118}$ iso flies. Considering non-significant differences in survival that we observed between different wild-type genetic backgrounds, we decided to use $w^{1118}$ iso for the experiments since most of the fly

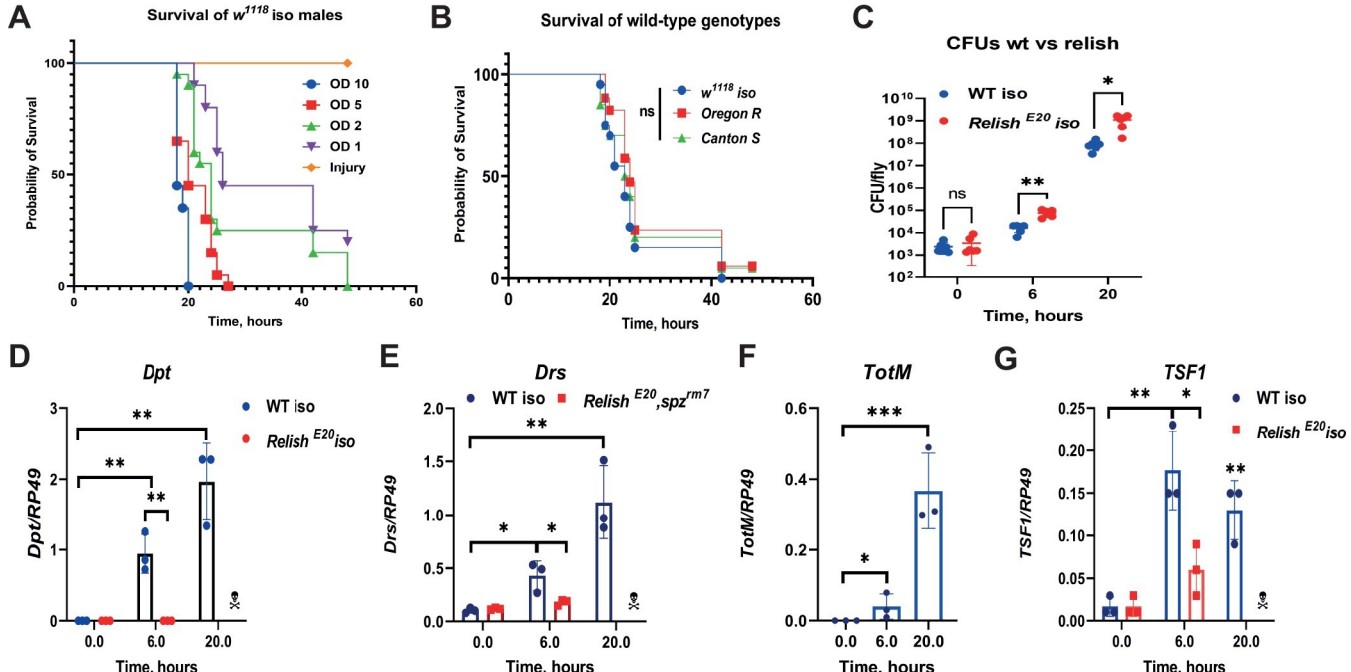

**Fig 1. *P. alcalifaciens* rapidly kills flies despite triggering immune response.** (**A**) Dose-dependent effect of *Pa* on $w^{1118}$ iso survival. (**B**) *Pa* is equally virulent to different wild-type fly genotypes. (**C**) *Pa* load significantly increases in WT and *Relish* flies over time. (**D-G**) RT-qPCR showing that *Pa* significantly induces expression of *Dpt* (**D**), *Drs* (**E**), *TotM* (**F**), *Tsf1* (**G**) in Imd (D, G) or Toll-dependent (E) manner. Results (C-G) are shown as mean ± SD of at least three independent samples.

mutants that we are using are in this background. Consistent with the high mortality of flies, we found that *Pa* proliferates rapidly inside flies and reaches up to $10^8$ CFUs per fly as early as 20h post infection (Fig 1C). Next, we wondered whether the host does not recognize *Pa* and therefore fails to mount an appropriate immune response, resulting in rapid proliferation of bacteria and death of the host. We measured *Dpt* expression as a read out of Imd pathway activation and found that *Pa* potently induced *Dpt* expression at 6 and 20h post infection in wild-type but not in *Relish* flies (Fig 1D). We also, found that *Pa* induces another two humoral immune pathways: Toll and JAK-STAT, as illustrated by *Drs* and *Turandot M* (*TotM*) expression (Fig 1E and 1F). Additionally, we detected induced expression of *Transferrin 1* (*Tsf1*) (Fig 1G)–a major player in infection-induced iron sequestration. Taken together, these results show that flies recognize *Pa* infection and trigger several immune pathways, but these responses are not sufficient to protect the flies from infection.

### *D. melanogaster* transcriptional response to *Pa* infection

Next, we used RNA-seq to obtain a system-wide view of transcriptional responses triggered by *Pa* infection in *D. melanogaster*. The aim of this approach was to test the possibility that flies fail to trigger timely immune response against *Pa*, thus rapidly dying from uncontrolled bacterial infection. For this experiment, we used $w^{1118}$ iso female flies at early (6 h) and late (20 h) timepoints after infection. The lists of differentially expressed genes are shown in S1 Table. At 6 h post infection, we detected 72 differentially-expressed genes, with 65 being induced and 7 repressed (Fig 2A). Long non-coding RNA IBIN (Induced by Infection) previously shown to link metabolism and immune response was the most upregulated transcript [65]. The top induced genes are dominated by the antimicrobial peptides regulated by the IMD pathway, like *DptA*, *DptB*, *AttB*, *CecA1*, *CecC*, *CecB*, *Dro*, consistent with a role of the Imd pathway in the defense against Gram-negative bacteria. AMPs and other genes regulated by the Toll pathway, namely *Drs*, *SPE*, *Bomanins*, *Daishos* [21] were also induced by *Pa*. This result suggests that the Toll pathway, typically induced by Gram-positive bacteria, is also induced by *Pa*, confirming the previous observation [62] that the differences in Toll and Imd pathway activation are quantitative and not qualitative or binary. *Tsf1* transcripts, which are regulated by both the Toll and the Imd pathways, [15] was also induced, suggesting that iron sequestration is triggered by *Pa* infection. At 20h post infection, 1019 genes were differentially expressed, with almost equal numbers of induced and repressed genes (Fig 2B). At this time point, again IBIN and Imd-regulated AMPs showed the strongest induction. We found that almost all genes (58 out 65) induced at 6h were also induced at 20h (Fig 2C). The overlapping genes (S1 Table) were mostly represented by AMPs, PGRPs, and signalling molecules of immune pathways. Gene ontology analysis of the upregulated genes showed enrichment of GO terms related to defense response, immune response, and antimicrobial humoral response, suggesting that most of the genes induced by *Pa* infection are involved in these processes (Fig 2D and 2E). Downregulated genes were associated with GO terms related to proteolysis, transmembrane transport and oxidation-reduction process, indicating the enrichment of metabolism-related genes (Fig 2F). Our transcriptomic analysis shows that *Drosophila* activates broad defense responses against *Pa* infection that are mainly regulated by Imd, Toll, and JAK-Stat signaling pathways, suggesting that flies fail to control infection despite induced immune responses.

### Imd pathway and iron sequestration play a role in the defense against *Pa* infection

Considering that we detected activation of major immune pathways after *Pa* infection, we asked whether they contribute to host protection. We assessed the survival of the respective

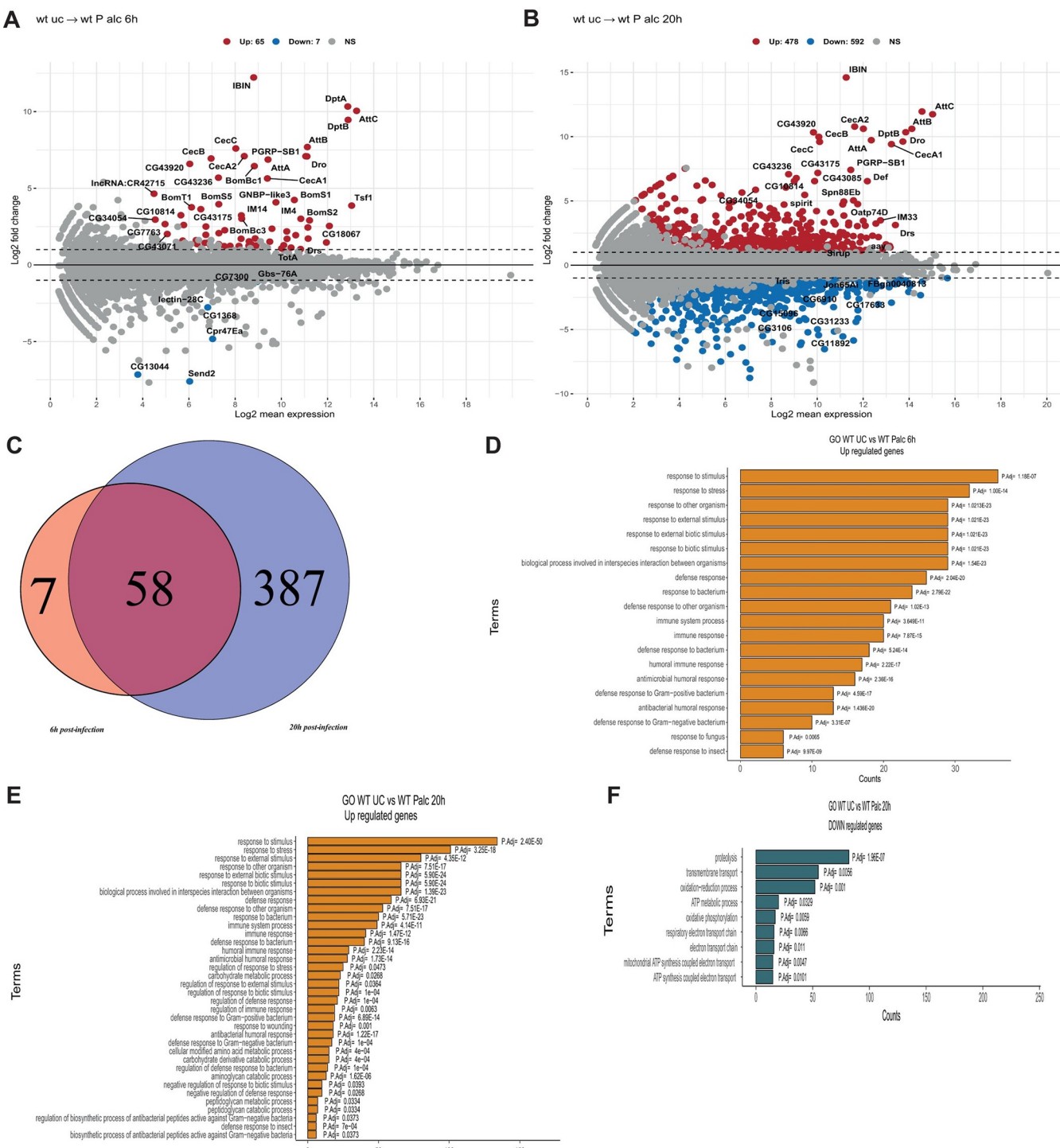

**Fig 2. Transcriptional changes after *Pa* infection.** MA plot of the *Drosophila* transcriptome showing log2 Fold change expression in wild-type flies ($w^{1118}$ iso) infected with *Providencia alcalifaciens*/ unchallenged, (x-axis) and (y-axis) after 6h (**A**) and 20h (**B**). Red and blue dots indicated differentially upregulated and down-regulated transcripts, respectively. (**C**) Venn diagram showing overlap between upregulated genes at 6h and 20h post infection. (**D-F**) Bar plots showing the enrichment of GO terms of global biological properties with significant gene counts upregulated at 6h (**D**) and 20h (**E**). (**F**) blue bar plots showing GO terms with down-regulated gene counts at 20h. P-adjusted values next to the bars.

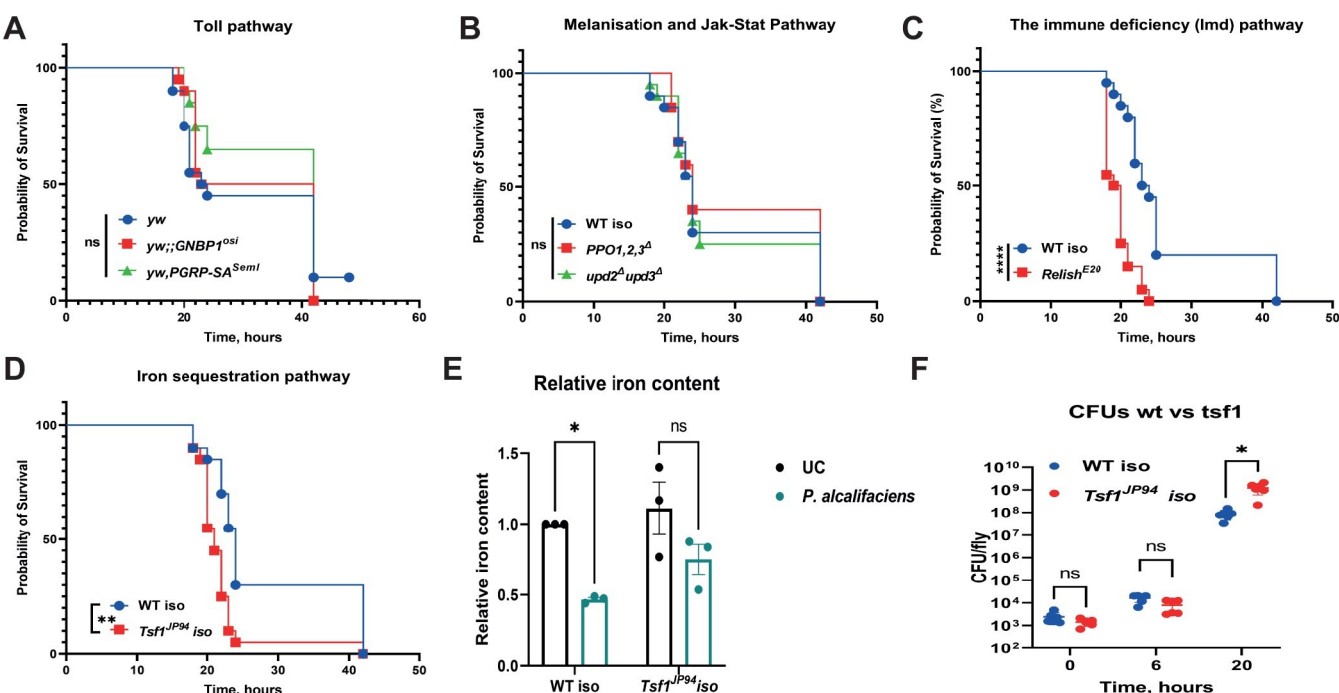

**Fig 3. Effect of major defense pathways on susceptibility to *Pa* infection.** (**A-D**) Survival rates of mutants in the following pathways: Toll (**A**), melanisation and Jak-Stat (**B**), Imd (**C**), iron sequestration (**D**) after infection with *Pa*. (**E**) Iron content assessed using ferrozine assay in the hemolymph of WT and *Tsf1* mutant flies 16h after *Pa* infection. (**F**) *Pa* reaches higher load in *Tsf1* mutant compared to WT flies 20h post infection. Results (E-F) are shown as mean ± SD of at least three independent samples.

mutants to *Pa* infection and found that flies lacking pattern recognition receptors of the Toll pathway, PGRP-SA and GNBP1, and a ligand of the Toll receptor, spaetzle, had similar survival (Figs 3A and S1A) and pathogen load (S1B Fig) to genetic background controls. In a similar way we could exclude the contribution of melanisation and the JAK-STAT pathway to fly protection from *Pa*, as flies lacking melanisation (*PPO1,2,3*) and cytokines of the Jak-Stat pathway (Δ*upd2-3*) survived *Pa* infection at the same rate as wild-type flies (Fig 3B) and carried the same pathogen load as wild type flies (S1C Fig). Flies lacking *Relish*, which are deficient in Imd pathway activation, were significantly more susceptible to *Pa* (Fig 3C) and had a higher pathogen load compared to wild-type flies (Fig 1C). This is consistent with a general role of the Imd pathway in the defense against Gram-negative bacteria. These results lead to the conclusion that although several immune pathways were induced after infection, only Imd appeared to have a relevant role, indicating that the genes that are transcriptionally responsive to a treatment are not necessarily the genes that are driving the physiological response to the treatment. Flies lacking *Tsf1* and thus infection-induced iron sequestration were also highly susceptible to *Pa* infection and died significantly faster than wild-type flies (Fig 3D). Consistent with previous findings with different pathogens [15], we observed a significant reduction in hemolymph iron level in wild-type flies but not in *Tsf1* mutant flies after *Pa* infection (Fig 3E). This excess of iron in *Tsf1* mutant flies, likely boosts *Pa* growth and consequently leads to fly death. Indeed, *Pa* reached higher loads in *Tsf1* mutant flies as illustrated by CFU estimation (Fig 3F). Altogether, our mutant survival analysis showed that the Imd pathway and iron sequestration contribute to the defense of *Drosophila* against *Pa* infection. However, these responses are not sufficient to control the pathogen, which likely has ways to evade them.

## Screen for attenuated *Pa* mutants

After characterizing the host responses to *Pa* infection, we turned our attention to the virulence mechanisms of the pathogen in hopes of understanding how *Pa* kills flies and evades the host defenses. Considering the limited prior knowledge on *Providencia* virulence, we decided to take an unbiased forward genetics approach. For this purpose, we generated a transposon mutant library consisting of around 3000 *Pa* mutants with random insertions. We screened this library for attenuated mutants by infecting flies with each individual mutant. We used 15 flies per *Pa* mutant and scored survivals for five days. Our initial screen identified 75 mutants with reduced virulence. After testing these mutants a second time and filtering out those affecting the same gene, we obtained 23 confirmed mutants attenuated in wild-type flies (Table 1). As shown in the survival graphs (Fig 4A–4W), they have different degrees of virulence: avirulent (PL1C4, PL14C2), strongly attenuated (PL11H9, PL2D4), and moderately virulent (PL13C10, PL1A3, PL4H7). The fitness of these mutants was not generally affected, as the vast majority showed similar growth in LB broth (S2A Fig). Only a few of these mutants (PL2D4, PL4B5, PL13H10) showed significantly reduced growth compared to wild-type bacteria (S2B Fig). When we examined the identity of the attenuated mutants, the vast majority of them (15 out of 23) were found to have the transposon insertions in genes associated with LPS synthesis or modifications (Table 1).

Since this group of mutants was the most common, we decided to focus further on them. First, we analyzed the LPS profile of some of the mutants by SDS-PAGE to see if we could detect differences compared to wild-type bacteria. Indeed, we found that mutants PL5D5 (UDP-N-acetylglucosamine diphosphorylase), PL3H8 (WxcM-like domain-containing protein), PL8A10 (O-antigen ligase), and PL8A11 (aminotransferase) displayed similar profiles with only two intense bands in the lipid A-core region instead of the four found in the WT strain (Fig 5). These mutants also displayed reduced O-chain signals, which is consistent with the annotated function of these genes in O-chain biosynthesis. Mutants PL11B8 and PL1C4 showed lipid A-core bands with lower molecular weight compared to the WT (Fig 5). This suggests that in both of these mutants the core region is affected which could have lost some sugar moieties and/or lipid A components. Indeed, both PL11B8 (lipopolysaccharide 1,3-galactosyltransferase) and PL1C4 (ADP-heptose—LPS heptosyltransferase) are involved in the LPS core biosynthesis based on studies in other bacteria [66]. Mutants P13H10 (OmpA), PL5A4 (YbaY family lipoprotein), P4G7 (glucose-1-phosphate-thymidylyltransferase) displayed similar profiles at both the core region and O-chain level compared to WT (Fig 5). This could imply that these mutations either do not affect LPS structure or they target very discreet modifications that cannot be detected with the method we used.

Next, we investigated how the identified mutations lead to reduced pathogen virulence. We hypothesized that altered LPS makes the bacteria more visible to the flies' immune system, so that they are better recognized by PRRs, elicit a stronger immune response, and are cleared fast after infection. To test this hypothesis, we measured *Dpt* expression as a readout of Imd pathway activity in wild-type flies after infections with *Pa* mutants. The vast majority of the mutants (14/23) triggered the same level of Imd activity as wild-type bacteria, 4 out 23 mutants showed lower *Dpt* expression, while only 5 out 23 mutants induced a stronger Imd activation (S3 Fig). Overall, the majority of attenuated mutants were as immunogenic as wild-type bacteria, suggesting that their reduced virulence was not due to the induction of a stronger immune response.

Alternatively, LPS modifications could interact with immune effectors of flies, like AMPs. Indeed, there is plenty of evidence that LPS is involved in such interactions by making bacteria more sensitive to AMPs [67]. Four families of *Drosophila* AMPs have been implicated in killing

**Table 1. Summary of the phenotypes of 23 attenuated *Pa* mutants.**

| Mutant name | Mutant identity | Sensitive to polymyxin | Sensitive to H2O2 | Rescued in *Relish* mutant | Rescued in *ΔAMP* mutant | *Rel* mutant rescued by *Drosocin* oe | Rescued by NAC | Rescued in *Duox* RNAi | Rescued in *Sod1* oe |
|---|---|---|---|---|---|---|---|---|---|
| PL1 C4 | ADP-heptose—LPS heptosyltransferase, RfaF | + | 0 | + | + | - | NA | NA | NA |
| PL2 D4 | Tol-Pal system protein, TolB | + | + | + | + | + | NA | NA | NA |
| PL3 H8 | WxcM-like domain-containing protein, O-Antigen gene cluster | + | 0 | + | + | + | NA | NA | NA |
| PL4 B5 | peptidoglycan-associated lipoprotein, pal | + | 0 | + | + | + | NA | NA | NA |
| PL4 E6 | hypothetical protein | + | 0 | + | + | + | NA | NA | NA |
| PL4 H7 | Proline-specific permease, ProY | + | + | + | + | + | NA | NA | NA |
| PL4 G7 | glucose-1-phosphate-thymidylyltransferase, rlmA, O-antigen gene cluster | + | 0 | + | + | + | NA | NA | NA |
| PL5 A4 | YbaY family lipoprotein | + | + | + | + | + | NA | NA | NA |
| PL5 D5 | UDP-N-acetylglucosamine diphosphorylase, O-antigen biosynthesis gene cluster | + | - | + | + | + | NA | NA | NA |
| PL7 D11 | Lipoprotein NlpI | + | + | + | + | + | NA | NA | NA |
| PL8 A10 | O-antigen ligase, rfaL | + | 0 | + | + | + | NA | NA | NA |
| PL8 A11 | aminotransferase, fdtC, O-antigen biosynthesis gene cluster | + | 0 | + | + | + | NA | NA | NA |
| PL 10 C8 | dTDP-3-amino-3,6-dideoxy-alpha-D-galactopyranose transaminase, fdtB (O-antigen biosynthesis gene cluster) | + | 0 | + | + | + | NA | NA | NA |
| PL 11 B8 | lipopolysaccharide 1,3-galactosyltransferase, rfaJ | + | + | + | + | + | NA | NA | NA |
| PL 11 H8 | hypothetical protein | + | 0 | + | + | - | NA | NA | NA |
| PL 11 H9 | Bifunctional polymyxin resistance protein, ArnA | + | 0 | + | + | + | NA | NA | NA |
| PL 13 H10 | porin OmpA | + | + | + | + | + | NA | NA | NA |
| PL 13 C10 | Sigma-E factor regulatory protein, rseB | 0 | 0 | + | + | - | NA | NA | NA |
| PL1 A3 | cytochrome o ubiquinol oxidase subunit III, cyoC | 0 | + | - | - | NA | + | + | + |
| PL4 F11 | Dihydrolipoyl dehydrogenase, lpdA | + | + | - | - | NA | + | + | + |
| PL6 B7 | GTP-binding protein TypA/BipA | 0 | + | - | - | NA | + | + | + |
| PL6 D10 | hypothetical protein | 0 | + | - | - | NA | + | + | + |
| PL 14 C2 | ATPase RavA stimulator ViaA | 0 | + | - | - | NA | ND | ND | ND |

This table list 23 *Pa* mutants that are significantly less virulent to wild-type flies (see Fig 4 for survival comparisons).

In polymyxin B and H2O2 sensitivity columns the following is shown:

+–mutant is more sensitive

–mutant is more resistant

0 –no change compared to wild-type *Pa*

In the other columns the following is applicable:

+–rescue (the virulence of this *Pa* mutant is significantly increased under this condition)

–no rescue (the virulence of this *Pa* mutant doesn't change under this condition)

ND–not done for this mutant

NA–not applicable to this mutant since other assays suggest a different mechanism

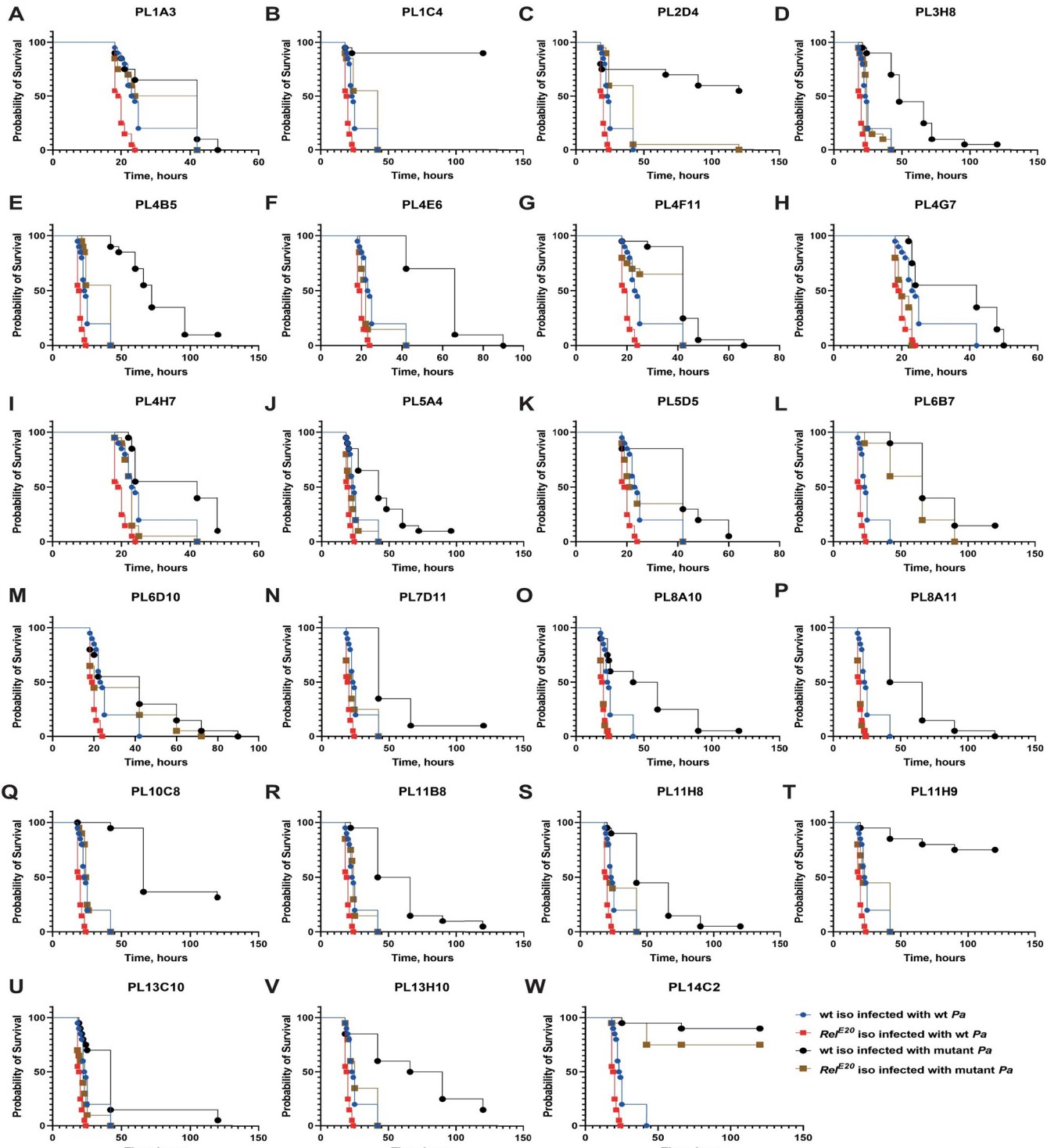

**Fig 4. *Pa* attenuated mutants differ in the virulence levels. (A-W)** Survival rates of 23 attenuated *Pa* mutants in WT and *Relish* flies. Vast majority of attenuated mutants retain their virulence in *Relish* flies. Mutants in (A, G, L, M, W) remain attenuated in *Relish* flies. See S3 Table for statistical comparisons.

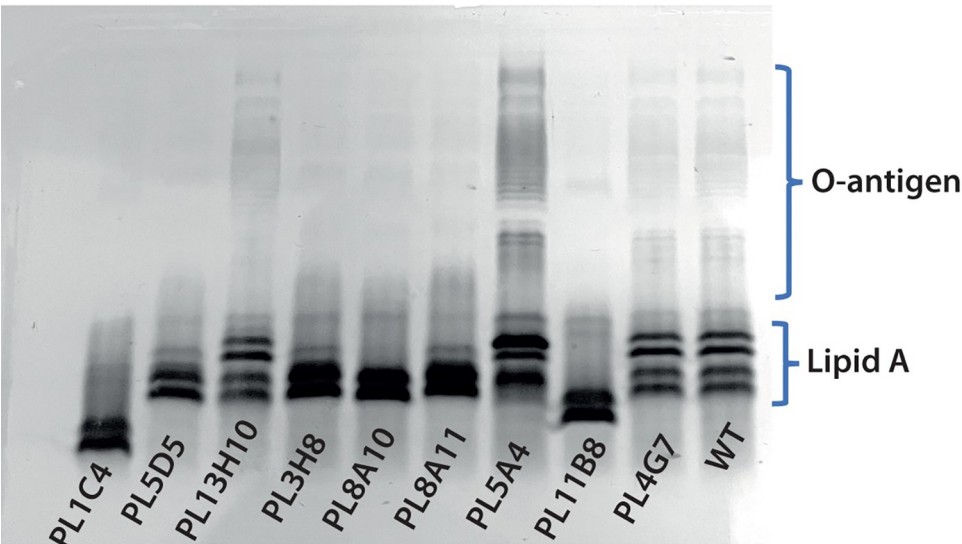

**Fig 5. SDS-PAGE profile of LPS from whole-cell samples.** LPS profile of WT *Pa* and of 9 attenuated *Pa* mutants is shown. Mutants PL5D5, PL3H8, PL8A10, and PL8A11 display only 2 intense bands in the lipid A region instead of 4 for the WT. These mutants also display reduced O-chain signals. Mutants PL11B8 and PL1C4 show lipid A-core bands at lower molecular weight compared to the WT. Mutants P13H10, PL5A4, P4G7 display similar profiles compared to WT both at the core and O-chain level.

Gram-negative microbes—Attacin, Cecropin, Diptericin, and Drosocin [68]. Since it is not possible to reproduce *in vitro* the full range of AMPs produced *in vivo* in response to an infection, we tested several individual AMPs for their effects on *Pa*. We used commercially available Cecropin A, Cecropin B, and synthesized on our request Diptericin B. However, none of the tested AMPs showed activity against *Pa* in a standard MIC assay. Bacterial growth was not inhibited even at the highest concentration (1mg/ml) that could be tested (S2 Table), while kanamycin used as a positive control effectively suppressed the growth at 12.5 μg/ml. Therefore, we decided to test the cationic antimicrobial peptide polymyxin B, which is frequently used to model AMP sensitivity [69,70]. We grew all *Pa* LPS mutants in the presence of polymyxin B and evaluated their growth by measuring OD600 over time. Both growth kinetics (Fig 6A) and the endpoint OD600 (Fig 6B) showed that 14 out of 15 tested mutants are more sensitive to polymyxin B compared to wild-type *Pa*. Next, we investigated whether this increased susceptibility to polymyxin B *in vitro* also leads to increased susceptibility to *Drosophila* AMPs produced *in vivo*. For this purpose, we tested the virulence of the polymyxin B-sensitive mutants on *Relish* mutant flies lacking Imd-dependent inducible AMPs expression. Survival showed that the virulence of all 15 *Pa* mutants was rescued in the *Relish* mutant background almost to the level of wild-type *Pa* (Fig 4B–4E, 4H, 4J, 4K, 4N, 4O, 4P, 4Q, 4R, 4T, 4W and 4U). Similarly, the growth of these mutants within the host was restored in *Relish* mutant flies when measured at 20h post infection (Fig 6C). Additionally, we compared the within host growth kinetics of wild-type *Pa* and 2 representative LPS mutants, PL10C8 and PL11H9, in wild-type and *Relish* mutant flies. While wild-type *Pa* proliferated monotonically in both wild-type and *Relish* mutant flies (S4A Fig), the burdens of PL10C8 and PL11H9 mutants did not increase and stayed rather stable in wild-type flies (S4B and S4C Fig), indicative of pathogen control by the host. However, both PL10C8 and PL11H9 mutants proliferated with similar kinetics to wild-type *Pa* in *Relish* mutant flies (S4B and S4C Fig), proving that the IMD pathway restricts these mutants in wild-type flies. Taken together, these results suggest that LPS

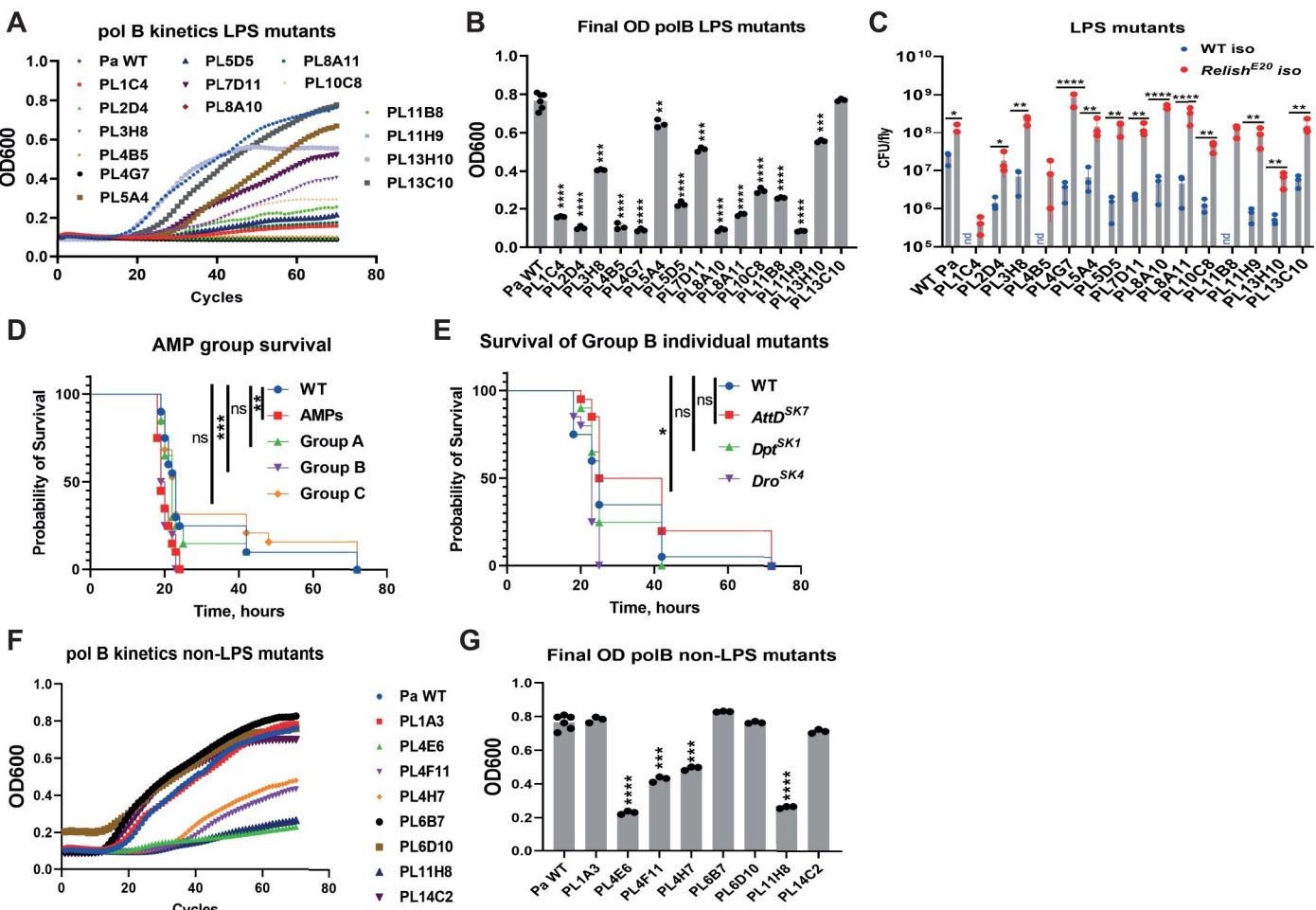

**Fig 6. *Pa* LPS mutants are sensitive to AMPs *in vitro* and *in vivo*. (A, B)** Sensitivity to polymyxin B of 15 *Pa* LPS mutants shown as growth kinetics (**A**) and as OD600 at the end of incubation (**B**) in LB containing polymyxin B. Asterisks indicate significance relative to *Pa* WT. All mutants but one (PL13C10) are more sensitive to polymyxin B compared to *Pa* WT. (**C**) *Pa* LPS mutants load in WT and *Relish* flies 20 h post infection. All mutants reached significantly higher load in *Relish* compared to WT flies. nd-not detected (likely below detection limit). (**D**) Flies lacking group B AMPs are as sensitive to *Pa* as *ΔAMP* flies. (**E**) Drosocin-deficient flies are hyper-sensitive to *Pa* infection. (**F, G**) Sensitivity to polymyxin B of 8 remaining *Pa* mutants with no known role in LPS shown as growth kinetics (**F**) and as OD600 at the end of incubation (**G**) in LB containing polymyxin B. Asterisks indicate significance relative to *Pa* WT.

mutants do not proliferate in wild-type flies because they are sensitive to the immune response mediated by the host IMD pathway, rather than because growth in the fly is impaired or there is a general fitness defect (consistent with in vitro results). Consequently, we wanted to identify the individual AMPs that were involved. We found that the *ΔAMP* mutant lacking 10 AMP genes [57] is more susceptible to wild-type *Pa* compared to wild-type flies (Fig 6D). Accordingly, the virulence of most LPS *Pa* mutants in the AMP mutant background was rescued (S5A–S5O Fig). Next, we utilized the group approach used by Hanson et al [57] to identify individual AMPs or a combination of AMPs that are essential for the defense against *Pa*. To this end, we infected three groups of combined mutants with wild-type *Pa*: A) flies lacking *Defensin* (Group A); B) Flies lacking three antibacterial and structurally related AMP families: the Proline-rich *Drosocin* and the Proline- and Glycine-rich *Diptericins* and *Attacins* (Group B) and C) Flies lacking the two antifungal peptide genes *Metchnikowin* and *Drosomycin* (Group C). Group A and group C flies survived *Pa* infection similar to wild-type flies, while group B flies showed increased susceptibility similar to the *ΔAMP* mutant (Fig 6D). In the next

step, we infected single group B AMP mutants and found that only flies lacking *Drosocin* are more sensitive to *Pa* infection (Fig 6E). Considering these results, we explored if overexpression of *Drosocin* in *Relish* flies would be sufficient to improve the resistance against *Pa* infection. In line with this, overexpression of *Drosocin* significantly increased the survival of *Relish* mutant flies after infection with a number of *Pa* LPS mutants, except PL13C10 and PL1C4 (S6A–S6O Fig). In contrast, overexpression of *Drosomycin* (S6P Fig) did not enhance the survival of *Relish* mutant flies. These results indicate that Drosocin particularly and not any AMP is important against *Pa* infection and that intact LPS is essential for bacteria to resist the action of this AMP.

Next, we decided to investigate whether the other attenuated mutants from our list, for example those, where hypothetical proteins are affected, might also be hyper sensitive to host AMPs. Indeed, for 4 out of 8 mutants we detected higher susceptibility to polymyxin B (Fig 6F and 6G) and 3 of them (PL4E6, PL4H7, PL11H8) could be rescued in *Relish* (Fig 4F, 4I and 4S) and *ΔAMP* (S5Q, S5S and S5V Fig) backgrounds, suggesting that these genes are required for resistance to host AMPs similar to the LPS mutants. Interestingly, mutant PL4F11 showed increased sensitivity to polymyxin B (Fig 6F and 6G), however its virulence was not rescued in *Relish* (Fig 4G) or *ΔAMP* mutants (S5R Fig), indicating that the reduced virulence of PL4F11 is not due to increased sensitivity to AMPs.

## Sensitizing *Pa* to host AMPs improves host survival

Our results identified LPS-mediated resistance to host AMPs as a prominent mechanism of *Pa* pathogenesis. We decided to exploit this mechanism for potential therapeutic applications. Namely, we tested if fly exposure to chemicals affecting LPS, like polymyxin B, could sensitize pathogens to host AMPs and improve the outcome of infection. For this purpose, we prefed flies with either sucrose (control) or a sucrose/polymyxin B mixture prior to *Pa* infection. Our survival analysis showed that flies that were preexposed to polymyxin survived *Pa* infection better compared to control flies (Fig 7). Importantly, this improved survival was not due to the

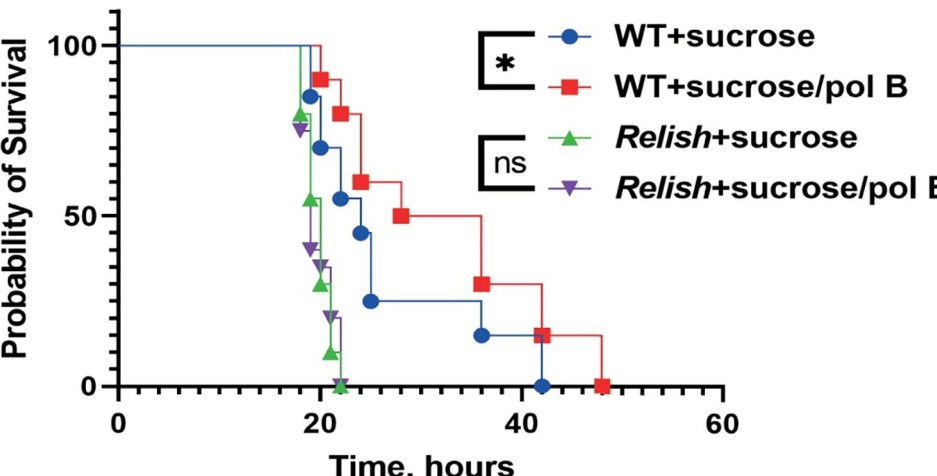

**Fig 7. Sensitizing *Pa* to AMPs by perturbing LPS improves host survival in Imd-dependent manner.** WT but not *Relish* flies that were fed on sucrose supplemented with LPS-targeting AMP polymyxin 24h prior to infection showed significantly improved survival after *Pa* infection.

direct antimicrobial activity of polymyxin B, as we used a concentration that was lower than what would be microbicidal and the *Relish* mutant flies did not show any improved survival after preexposure to polymyxin (Fig 7). This finding together with our other results suggests that polymyxin increases pathogen susceptibility to Imd-dependent AMPs likely by disrupting LPS of *Pa*, and is thus helping the host to eliminate the pathogen.

## ROS resistance as a virulence mechanism

Next, we decided to have a closer look at the remaining five mutants (PL1A3, PL4F11, PL6B7, PL6D10, PL14C2) that did not show increased susceptibility to polymyxin B (Fig 6F and 6G) (except PL4F11) and were not rescued in *Relish* (Fig 4A, 4G, 4L, 4M and 4W) and *ΔAMP* (S5P, S5R, S5T, S5U and S5W Fig) mutant background. Interestingly, some of the genes disrupted in these mutants (PL1A3, PL4F11, PL6B7) were previously linked to the reactive oxygen species (ROS) response [71]. Therefore, we tested the sensitivity of all attenuated mutants to ROS by using an H2O2 disk diffusion assay (S7 Fig). Indeed, PL1A3, PL4F11, PL6B7 and PL6D10, PL14C2 mutants were more susceptible to ROS (Fig 8A). Notably, some of the LPS mutants also showed increased susceptibility to ROS although to a lower degree, and one mutant PL5D5 was completely resistant (Fig 8A). To further investigate the relationship between ROS and *Pa* virulence, we decided to use the four mutants which are highly sensitive to ROS (PL1A3, PL4F11, PL6B7, PL6D10). We excluded PL14C2 from the further analysis since the low virulence of this mutant did not allow for a conclusive comparison between the treatments. ROS are immune effectors [7], so susceptibility to them might impair the ability of bacteria to survive inside the fly and consequently their virulence. Indeed, we could detect lower CFUs for two *Pa* ROS-sensitive mutants (PL6B7, PL6D10) in wild-type flies (Fig 8B). The growth within the host of the ROS-sensitive mutants in contrast to the LPS mutants was not rescued in *Relish* flies (Fig 8B), suggesting that a mechanism other than Imd activation restricts the proliferation of ROS-sensitive *Pa* mutants. To prove that ROS-sensitive *Pa* mutants are attenuated because of increased susceptibility to host ROS, we prefed flies with the ROS scavenger NAC prior to infection, and observed that ROS-susceptible mutants but not wild-type *Pa* were more virulent in NAC-treated flies (Fig 8C–8F). Thus, ROS scavenging can rescue the attenuated phenotype of the mutants. Next, we investigated the source of ROS during *Pa* systemic infection. The melanization reaction has been proposed to trigger oxidative stress in the case of *Drosophila* infection with *Francisella novicida* [70]. Therefore, we tested the survival of ROS sensitive *Pa* mutants in *PPO1,2* flies lacking melanisation. However, we could not see any rescue of *Pa* virulence in flies lacking melanisation, suggesting that melanisation is not the source of ROS during *Pa* infection (S8A–S8D Fig). Next, we tested whether hemocyte-derived ROS contributes to the defense against *Pa* [72]. Indeed, genetic ablation of hemocytes made flies susceptible to ROS-sensitive *Pa* mutants (S8E–S8I Fig). To prove that hemocyte-deficient flies are more susceptible due to the lack of ROS rather than phagocytosis, we knocked-down *Duox*—a major ROS generating enzyme in hemocytes [72]. Genetic ablation of *Duox* specifically in hemocytes had a small but non-significant effect on the susceptibility of flies to wild-type *Pa*, however markedly increased susceptibility to ROS-sensitive *Pa* mutants (Fig 8G–8K). Additionally, we tested the overexpression of ROS scavengers—extracellular *Sod1* and intracellular mitochondrial superoxide dismutase *Sod2* [73]. Flies overexpressing *Sod1* but not *Sod2* displayed increased susceptibility to ROS-sensitive *Pa* mutants (Fig 8G–8K). Overall, these data suggest that hemocyte-derived, Duox-mediated ROS are important defense molecules against *Pa* infection in flies. The ability of *Pa* to resist these ROS is an essential virulence strategy.

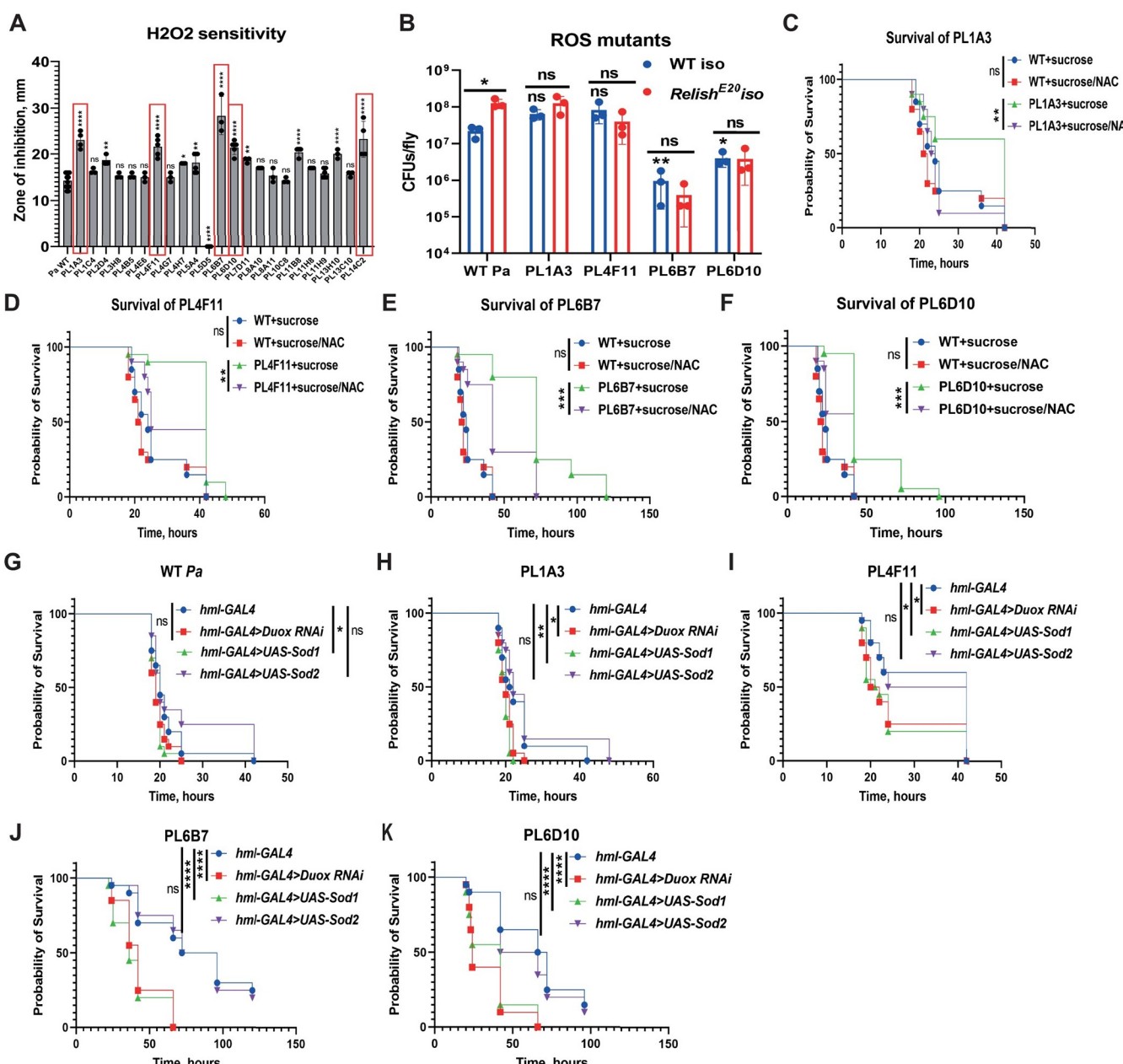

**Fig 8. Resistance to hemocyte-produced ROS is part of *Pa* virulence mechanism.** (**A**) Sensitivity to H2O2 of 23 attenuated *Pa* mutants measured by disc-diffusion assay shows that 5 *Pa* mutants (in red) are particularly sensitive to ROS. These mutants are not sensitive to polymyxin and not rescued in *Relish* mutant, suggesting that their phenotype is independent of host Imd pathway. (**B**) *Pa* mutants load in WT and *Relish* flies 20 h post infection. Two mutants reached lower load in WT flies, while none of the mutants grew better in *Relish* flies. Asterisks or ns above blue bars indicate significance relative to *Pa* WT in WT flies. (**C-F**) Virulence of ROS-sensitive *Pa* mutants is significantly increased in flies fed sucrose/NAC mixture prior to infection to scavenge ROS. (**G-K**) Knock-down of ROS-producing enzyme *Duox* and overexpression of ROS-scavenging enzyme *Sod1* specifically in hemocytes increases susceptibility to ROS-sensitive *Pa* mutants.

## Discussion

The aim of our study was to dissect the host-pathogen interactions between *Providencia* and *D. melanogaster*. To achieve this aim, we utilized various genetic approaches that enabled us to determine the contributions of both pathogen and host to the outcome of the infection. First,

we characterized the responses of the fruit fly to *Pa* infection and, using mutant analysis, identified the Imd pathway and iron sequestration as prominent defense mechanisms against *Pa*. Second, we performed an unbiased forward genetics screen using a transposon mutant library that we generated for this purpose and identified *Pa* virulence factors necessary to infect the fly. This mutant library has the potential to serve as a valuable resource for exploring the genetic basis for all *Pa* traits. Third, we used mutants of the major immune pathways in *Drosophila* and infected them with attenuated *Pa* mutants to identify pathogen virulence factors that allow the bacteria to respond to specific immune defenses and evade immune clearance. Thereby, we dissected both sides of host-pathogen relationship in a *Drosophila-Providencia* model and provided the first insights into the molecular mechanisms of *Pa* virulence.

To identify *Pa* virulence factors, we performed an *in vivo* screen which yielded 23 attenuated mutants. The majority of these mutants (15/23) had transposon insertions in genes involved in LPS biosynthesis and LPS modifications, pointing towards a vital role of intact LPS in *Pa* pathogenesis. This finding is consistent with a well-known role of LPS in host-pathogen interactions [67,74]. At the mechanistic level, LPS protects *Pa* from *Drosophila* Imd pathway-dependent AMPs, particularly Drosocin. Consistent with this, *Pa* LPS mutants showed increased susceptibility *in vitro* to the cationic AMP polymyxin B and their virulence was restored in *Relish* and *ΔAMP* mutant flies deficient for Imd-dependent AMPs. Our finding that *Pa* LPS mediates resistance to host AMPs complements numerous previous studies in diverse pathogens that reported a similar protective function of LPS against host innate defenses [67,75–78]. Several studies that used *Drosophila* as an infection model also discovered LPS as an essential protective barrier against insect AMPs. For example, Nehme et al found that LPS O-antigen-deficient *Serratia marcescens* mutants were attenuated in wild-type flies but not in an Imd pathway mutant [29]. A similar phenotype was reported for *F. novicida* mutants with affected LPS [70,79]. These data demonstrate that a major determinant of virulence in several pathogens is the LPS-mediated ability to resist the systemic immune response. Additionally, LPS was shown to facilitate microbiota-host interactions. For instance, LPS biosynthesis mutants of *Acetobacter fabarum*, a *Drosophila* commensal, had a reduced ability to colonize the fruit fly intestine [80]. While the mechanism behind this phenotype has not been investigated yet, increased sensitivity to intestinal AMPs is a likely reason, as shown for the human commensal *Bacteroides thetaiotaomicron* [81]. Among our LPS mutants, we found ArnA (pmrA) (PL11H9) which encodes an enzyme that catalyzes the formation of modified arabinose UDP-L-4-formamido-arabinose (UDP-L-Ara4FN). The modified arabinose reduces the negative charge of lipid A and the binding of cationic AMPs [69]. This is the most commonly observed LPS modification implicated in cationic AMP resistance. This modification is also crucial for *Yersinia pestis* resistance to the insect cecropin-like AMP cheopin [82].

In addition to mutations affecting LPS, we uncovered several that disrupt lipoproteins, like OmpA (PL13H10), NlpI (PL7D10), and YbaY (PL5A4). While YbaY is poorly characterized, OmpA and NlpI were previously implicated in the virulence of different pathogens [83,84]. Whereas OmpA contributes to virulence in various ways ranging from facilitating adhesion and invasion to conferring resistance to serum [83], NlpI function in virulence is less clear. Our results suggest that all three lipoproteins mutants behave like LPS mutants–they are susceptible to polymyxin B and their virulence is rescued in an AMP mutant, indicating that their reduced virulence is due to an increased susceptibility to host AMPs. The mechanism behind this phenotype requires further investigation, however NlpI was shown to be essential for cell envelop integrity [85], which might contribute to increased sensitivity to AMPs. Our screen uncovered two additional peptidoglycan-associated lipoproteins, TolB (PL2D4) and Pal (PL4B5), that are part of a multiprotein complex, the Tol-Pal system. It bridges between the peptidoglycan and the outer membrane and is important for proper structure and function of

the outer membrane [86]. Importantly, TolA and Pal are necessary for correct surface polymerization of O-antigen chains, likely explaining the sensitivity of *tol* and *pal* mutants to detergents and several antibiotics [87]. Similar to our *Pa tol* and *pal* mutants, *F. novicida* mutants in these genes were attenuated in *Drosophila* infection and more sensitive to host AMPs [79].

The second largest group of mutants with reduced virulence that we identified constitutes ROS-sensitive mutants. Since we could rescue the virulence of these mutants by chemical or genetic ROS scavenging, their attenuated virulence is likely due to an inability to resist host ROS produced in response to infection. Among such ROS-sensitive mutants, only the one lacking cytochrome oxidase (PL1A3) was previously shown to be required for virulence in other bacteria by enhancing the tolerance to oxidative stress [71]. Some other genes, like *dihydrolipoyl dehydrogenase* (PL4F11) and *typA* (PL6B7), were also linked to virulence but not necessarily via ROS sensitivity [88,89]. We could not find any previous evidence of the role of ATPase RavA stimulator ViaA (PL14C2) in virulence, however there seems to be a link to ROS response in *E. coli* [90]. Therefore, further investigation of the identified genes is required to clarify their role in bacterial virulence and ROS sensitivity. While previous studies identified several sources of ROS in flies, including melanisation [70,91], hemocytes [72,92], Nox [93,94] and, Duox [95], our results showed that Duox specifically in hemocytes is the major producer of ROS in case of *Pa* infection. Notably, in case of *F. novicida* melanisation played a prominent role as a source of ROS [70]. An interesting avenue for future studies would be to understand the differences between Duox- and melanisation-derived ROS and their preferential activity against specific pathogens.

Our screen also identified several hypothetical proteins. Using ROS and polymyxin B sensitivity assays and rescue in AMP- and ROS-deficient flies, we showed that PL4E6 and PL11H8 contribute to bacterial resistance to host AMPs, while PL6D10 is necessary to survive ROS exposure. Thus, with our approach we could assign a mechanism of virulence to hypothetical proteins with unknown function. However, how those protein contribute to ROS or AMP sensitivity remains unknown.

We identified one *Pa* mutant (Sigma-E factor regulatory protein rseB, PL13C10), that was not sensitive to ROS and polymyxin *in vitro*. However, the virulence of this mutant was rescued in *Relish* and AMP-deficient flies. Very likely the rescue phenotype could be due to sensitivity to additional antimicrobial peptides produced by flies. Such increased sensitivity to AMPs is possible given the role of Sigma-E factor in cell envelope integrity [96].

Among all AMPs tested, Drosocin proved to be particularly important in controlling *Pa* infection. Consistent with our *Pa* LPS mutants, *F. novicida* mutants in LPS were particularly sensitive to Drosocin [79]. Considering that Drosocin is known to bind bacterial LPS [97], alterations in LPS might promote Drosocin interactions with LPS and bacterial killing or make intracellular targets more accessible. A previous *in vivo* analysis of AMP specificity has shown that Drosocin plays a critical role in controlling *Enterobacter cloacae* infection [57]. A recent study confirmed this finding [56], however additionally reported that the *Drosocin* gene encodes not one, but two AMPs: Drosocin and IM7 (newly named as Buletin). Buletin but not Drosocin contributes to host defense against *Providencia burhodogranariea* infection. Since the *Drosocin* mutant we used lacked both Drosocin and Buletin and the *Drosocin* overexpression line similarly produced both peptides, it remains to be tested whether Drosocin or Buletin or both peptides together are involved in the defense against *Pa*.

While our in vivo experiments demonstrate that AMPs are the major Relish-regulated molecules controlling *Pa* LPS mutants, in vitro assays with synthesized *Drosophila* AMPs were not conclusive. None of the three *Drosophila* AMPs that we tested, Cecropin A, Cecropin B, and Diptericin B, showed activity against *Pa*. Considering the high specificity of some AMP-microbe interactions, it could be that the peptides we tested have no effect on *Pa*. Indeed, based on in vivo results, Drosocin, which was not available for an in vitro test, is the primary AMP controlling *Pa*

infection. Additionally, in vitro effects of AMPs can be different than in vivo effects of mutants or knockdowns for the same AMPs [98], suggesting that physiological context or interaction among peptides is important. Also, there are a number of technical reasons why in vitro assays may not reflect in vivo activities, including AMPs adhering to plastic assay plates, differences in salt concentrations or pH, stress on microbes, interactions among AMPs and between AMPs and other components of the immune system. These potential issues have to be considered when interpreting the results of in vitro antimicrobial tests performed with AMPs.

Contrary to our expectations, in our screen we did not hit any bacterial effectors, like toxins, that might be responsible for damaging the host. Since toxins are likely to be redundant, disruption of an individual toxin gene may not give a phenotype. Similarly, we did not identify any mutants in secretion systems, suggesting that *Pa* does not require effector translocation to infect *Drosophila*. The only toxin that was so far implicated in *Pa* pathogenesis is cytolethal distending toxin which blocks eukaryotic cell proliferation [99]. Interestingly, *Pa* LPS was shown to cause epithelial barrier dysfunction by reducing occludin levels in Caco-2 cell monolayers and induced apoptosis in calf pulmonary artery endothelial cells [100]. Thus, LPS might not only mediate resistance to host AMPs but also act as an effector-like molecule.

By discovering the mechanisms of *Pa* resistance to host AMPs and ROS, our study opens the doors to potential strategies to exploit such *Pa* mechanisms and sensitize the pathogen to host defenses to improve infection treatment. To illustrate the feasibility of such an approach, we used polymyxin B treatment to disrupt *Pa* LPS *in vivo* and found that it was sufficient to improve *Drosophila* survival after infection. Such beneficial effect of polymyxin B required functional Imd pathway signalling and was independent of direct bactericidal activity, suggesting that disruption of the major barrier against AMPs sensitizes the pathogen to host defenses. Our results suggest that affecting LPS function might be a useful strategy to treat *Providencia* infections, particularly those resistant to antibiotics.

Sensitizing *Pa* to host ROS also appears to be an attractive anti-virulence strategy [101], considering that resistance to host ROS is one of the key *Pa* virulence mechanisms that we identified. Some compounds were shown to sensitize the pathogens to oxidative stress and immune clearance but in a species-specific manner. For example, 2-[2-nitro-4-(trifluoromethyl) benzoyl]-1,3-cyclohexanedione (NTBC) treatment inhibits production of pyomelanin pigment and increases sensitivity of pyomelanogenic *Pseudomonas aeruginosa* strains to oxidative stress [102]. Similarly, BPH-642 –cholesterol biosynthesis inhibitor, blocked biosynthesis of staphyloxanthin antioxidant pigment in *S. aureus*, resulting in increased immune clearance in a mouse infection model [103]. However, to date there are no known compounds that would predispose *Pa* or generally any pathogen to ROS without being toxic to the host, thus limiting the development of ROS-potentiating anti-infectives.

In summary, our reciprocal analysis of interactions between *D. melanogaster* and *P. alcalifaciens* revealed that the host relies on Imd-dependent AMPs and hemocyte-derived ROS as major branches of immunity that are important for fighting infection with *P. alcalifaciens*. On the pathogen side, we have found that the ability to resist these host immune responses is the major virulence mechanism of *P. alcalifaciens*. Leveraging this knowledge has great potential to improve *P. alcalifaciens* infection treatment either by potentiating the host defenses or disrupting pathogen virulence.

## Materials and methods

### *Drosophila* stocks and rearing

The following *Drosophila* stocks used in this study were kindly provided by Bruno Lemaitre: DrosDel *w^1118* iso; Oregon R; Canton S; *Relish^E20* iso; *Relish^E20*, *spz^RM7* iso; *y,w, P{Dpt-lacZ,*

*ry+}, P{Drs-GFP, w+} (y,w,DD); ywDD;;GNBP1^{osi}; ywDD, PGRP-SA^{Seml}; PPO1^Δ,2^Δ,3^1; upd2, upd3^Δ; Tsf1^{IP94}* iso; *ΔAMP*; group A; group B; group C; *AttD^{SK7}; Dpt^{SK1}; Dro^{SK4}; UAS-bax; c564-GAL4; UAS-Drosocin, spatzle^{RM7}*. The following stocks were obtained from the Bloomington Drosophila Stock Center: *hml-GAL4, UAS-GFP* (30140); *Duox RNAi* (33975); *UAS-Sod1* (24750); *UAS-Sod2* (24494). The stocks were routinely maintained at 25˚C with 12/12 h dark/light cycles on a standard cornmeal-agar medium: 3.72g agar, 35.28g cornmeal, 35.28g inactivated dried yeast, 16 ml of a 10% solution of methyl- para- ben in 85% ethanol, 36 ml fruit juice, 2.9 ml 99% propionic acid for 600 ml. Fresh food was prepared weekly to avoid desiccation. Female flies were used in RNA-seq and qPCR experiments due to higher RNA yield, male flies were used in all other experiments.

## Bacterial strain, survival, and bacterial load

In this study, we used *Providencia alcalifaciens* DSM30120 obtained from German Collection of Microorganisms and Cell Cultures (DSMZ). The strain was routinely grown in LB media overnight at 37˚C with shaking. The culture was pelleted by centrifugation to concentrate microbes and diluted to the desired optical density (OD600 = 1, 2, 5, 10) with sterile PBS. To infect flies, a 0.15 mm minuten pin (Fine Science Tools) mounted on a metal holder was dipped into the diluted overnight culture and poked into the thorax of a CO2 anesthetized fly. Sterilely wounded flies were pricked with a needle that was sterilized in 95% ethanol. Infected flies were maintained in vials with food at 25˚C and surviving flies were counted at regular intervals (every hour from 18 to 24h post infection and every 24h after that) for 5 days after infection. Typically, 20 male flies per vial with 2–3 replicate experiments were performed. Survival experiments that involved prefeeding flies with chemical compounds involved overnight feeding of control group on 2.5% sucrose solution applied on top of a filter disk covering the fly food and experimental group feeding on a mix of sucrose+100 mM NAC or sucrose+10mg/ml polymyxin B.

For bacterial counts, flies were infected with *Pa* as described above, and the number of bacteria was determined as follows at 6 h and 20 h post-infection. Flies were surface sterilized in 95% ethanol for 1 min, and then 5 flies were homogenized in 500 μl of sterile PBS for 30 s at 6000 rpm using a Precellys 24 instrument (Bertin Technologies, France). Serial 10-fold dilutions were made and plated on LB culture medium using an automatic diluter and plater easy-Spiral Dilute (Interscience, France). The plates were left to dry and incubated overnight at 37˚C. Colonies were counted using an automatic colony counter Scan 1200 (Interscience, France) and its accompanying software.

## Generation of transposon mutant library

Commercially available EZ-Tn5 <R6Kyori/KAN-2>Tnp Transposome Kit (Lucigen) was used to generate a transposon library of *P. alcalifaciens* mutants. For this, electrocompetent cells of *P. alcalifaciens* were prepared by collecting the cells when they reached OD 0.5 and washing them 3 times with cold 10% glycerol. 100 μl of electrocompetent cells were mixed with 1 μl of transposome complex and placed in 0.2 mm electroporation cuvette (Biorad). The cells were electroporated using Gene Pulser Xcell System (Biorad) with the following settings 2.5 kv, 200 ohms and 25 μF. Cell were recovered immediately following the pulse with 1 ml of LB media and incubated for 1 h at 37˚C before plating on LB agar plates with 50 μg/ml kanamycin. Following overnight incubation at 37˚C, individual colonies were picked into the wells of 96 deep-well plates containing 1 ml of LB+kanamycin per well. The plates were covered with air-permeable sealing films and incubated at 37˚C overnight. Next day, 750 μl of 50% glycerol were added to each well and the plates were placed at -80˚C freezer for storage.

### Transposon screen and identification of insertion sites

Mutants were grown in 96 well plates under stationary conditions in LB+kan overnight. Next day, 15 $w^{1118}$ iso males were pricked per mutant using bacterial culture directly from the growth plate and survival was monitored for 2 days. Those mutants that killed flies at below average level were retested second time. For second retest, mutants were cultured in LB broth overnight with shaking, OD-adjusted (final OD 2), and compared to wild-type *Pa*. Those mutants that were confirmed to have reduced virulence in second retest were retained for transposon insertion site identification.

To identify transposon insertion sites, we used "rescue cloning" technique which is possible due to the R6Kγori present in the transposon. R6Kγori allows circular DNA to behave as a plasmid. Briefly, genomic DNA was extracted from *Pa* mutants of interests using Monarch Genomic DNA Purification Kit (NEB). This DNA was randomly fragmented with NEBNext dsDNA Fragmentase (NEB). Fragmented DNA was end-repaired (made blunt-ended) and 5′-phosphorylated using the End-It DNA End-Repair Kit (Lucigen). Finally, it was self-circulized using T4 DNA-Ligase (Thermofisher) and transformed into TransforMax EC100D *pir+ E. coli* (Lucigen) that expresses the pir gene product (the "pi" protein). When selected on kanamycin-containing plates, only the cells containing the <R6Kγori/KAN-2> Transposon can grow. Plasmids were extracted from the colonies using Monarch Plasmid Miniprep Kit (NEB) and used for sequencing of transposon-flanking DNA with transposon-specific primers supplied with the kit. Blast search of the obtained sequences was performed to determine the identity of the transposon-disrupted genes.

### RT-qPCR

For quantification of mRNA, 10 whole flies per sample were collected at indicated time points post infection into 1.5 ml screwcap tubes with 100 μl of glass beads. Total RNA was isolated using TRIzol reagent according to manufacturer's protocol and dissolved in RNase-free water. Quality of the RNA was determined using a NanoDrop ND-1000 spectrophotometer. Five hundred nanogram of total RNA was then reverse-transcribed in 10 μl reaction volume using PrimeScript RT (TAKARA) and random hexamer primers. Quantitative PCR was performed on a LightCycler 480 (Roche) in 384-well plates using the SYBR Select Master Mix from Applied Biosystems. Expression values were normalized to RP49. Primer sequences were published previously [104].

### RNA-seq and GO analysis

Total RNA was extracted from 10 whole flies per sample using TRIzol reagent. Total RNA was dissolved in nuclease-free water and RNA concentration was measured using a Nanodrop (Thermo Scientific). RNA integrity and quality were estimated using a Bioanalyzer (Agilent Technologies). Separate libraries for the three experimental conditions (flies infected with *Pa* 6h, 20h as well as uninfected controls) belonging to three independent experiments were prepared with the TruSeq RNA Sample Prep kit (Illumina, San Diego, CA) according to the manufacturer's protocol. Adapters containing seven nucleotide indexes were ligated to the double-stranded complementary DNA (cDNA). The DNA was purified between enzymatic reactions and the size selection of the library was performed with AMPure XT beads (Beckman Coulter Genomics, Danvers, MA). The libraries were pooled and sequenced using Illumina HiSeq 3000 instrument (75-bp paired-end sequencing) at the Max Planck-Genome-centre Cologne, Germany (https://mpgc.mpipz.mpg.de/home/).

RNA-seq data from this study (PRJNA833403) were analyzed with CLG Genomics Workbench (version 12.0 & CLC Genomics Server Version 11.0); Using the functions "Trim Reads"

[105] and "RNA-Seq Analysis." BDGP6.28 from the Ensembl was used for mapping and read counting as the reference genome. Differential expression analysis was done using DESeq2 [106]. Gene Ontology Analysis: Go term enrichment for gene group lists was conducted using FlyMine [107]. As background, the list of 12319 reproducibly measured genes was chosen. Results were filtered using a corrected p-value of <0.05 (Bonferroni). Data Visualization

The R packages ggplot2, dplyr, org.Dm.eg.db, were used for data visualization.

## Hemolymph extraction and Ferrozine-based colorimetric assay

Approximately 50 flies were infected with *P. alcalifaciens* ($OD_{600}$ = 2) and incubated for 16 ½ hours at 29°C. Hemolymph was extracted using Mobicol "Classic" spin columns (MoBiTec) in which two filters (large and small, 10 µm pore size) were inserted. Flies were added to the columns, which were then placed in collection tubes (containing 50µl protease inhibitor cocktail (Sigma Aldrich, one tablet in 4 ml PBS)), and glass beads were added to fill the column. The samples were then spun at 5000g for 10 min (at 4°C). Protocol adapted from [104].

Iron in the hemolymph of flies was quantified using a ferrozine-based colorimetric assay as described in [108] with minor modifications. The protein concentration in the hemolymph was measured using the Pierce BCA Protein Assay (Thermo Fisher Scientific) according to the manufacturer's protocol. A total of 120 µg protein was used for each condition and samples were adjusted to 50µl with the protease inhibitor cocktail. Samples were then treated as described in [108] and absorbance was measured at 562 nm using an infinite 200 Pro plate reader (Tecan). Quantification was performed using a standard curve generated with serial dilutions of a 10mM FAC stock dilution. The experiment was repeated three times.

## LPS analysis

Whole-cell samples were used for LPS analysis following previously published protocol [109]. The samples were resolved on 15% polyacrylamide gels with a 4% stacking gel. The LPS were stained using the Tsai and Frash method [110].

## Polymyxin B and ROS sensitivity

Sensitivity to polymyxin B was determined by assessing the growth of bacteria in LB medium containing 12.5 mg/ml of polymyxin B. This concentration, empirically-chosen, worked best to compare the sensitivities of different mutants. To set the assay, the optical density of all test strains was set to 0.1 in LB media containing polymyxin B. 150 µl of this suspension was pipetted in triplicates in wells of 96-well plate. The plate was incubated at 37°C in the plate reader Tecan Infinite 200 PRO which measured the optical density of each well every 20 minutes for 23 hours.

To determine the sensitivity of various *Pa* mutants to oxidative stress, disk diffusion assay was performed using protocol adapted from [70]. Briefly, 50µL of overnight cultures of bacteria were plated onto LB agar plates to create a lawn of bacteria. Plates were allowed to dry for 10 minutes, and then 6mm Whatman filter paper disks (Fisher Scientific) were placed onto each plate and inoculated with 10µL of 100mM freshly diluted hydrogen peroxide (Sigma). Plates were incubated overnight and the diameter of the zone of inhibition was measured for each sample. Three zones were measured for each mutant and each experiment was repeated three times.

## Statistical analysis

Data representation and statistical analysis were performed using GraphPad Prism 9 software. Survival curves were compared using log-rank tests, with Bonferroni corrections for *p* values

where multiple comparisons were necessary. All survival graphs show one representative experiment out of three independent repeats with 2–3 cohorts of 20 male flies per genotype. One-way ANOVA was used to analyze data in Figs 1F, 6B, 6G, 8A and S1B and S2. Two-way ANOVA was used to analyze data in Figs 1C, 1D, 1E, 1G, 3E-3F, 6C and 8B. Where multiple comparisons were necessary, appropriate Tukey, Dunnett, or Sidak post hoc tests were applied. Other details on statistical analysis can be found in Figure legends. Statistical significance was set at p≤0.05. Asterisks indicate *p≤0.05, **p≤0.01, ***p≤0.001, ****p≤0.0001, ns-non-significant, p>0.05.

## Supporting information

**S1 Fig. Toll and Jak-Stat pathways have no role in controlling *Pa* infection. (A)** Survival rates of *spaetzle* mutants and wild-type flies after infection with *Pa* are similar. **(B, C)** *Pa* reaches the same load in *GNBP1* (**B**) and *upd2,upd3* (**C**) mutants as in background control flies. Results (B-C) are shown as mean ± SD of at least 10 samples. Dots represent samples. Each sample included 5 flies and CFUs in each sample were normalized to the number of flies to obtain CFUs per single fly.
(TIF)

**S2 Fig. General fitness of attenuated *Pa* mutants in LB media. (A, B)** Growth kinetics of 23 *Pa* mutants in LB medium (**A**) and OD600 at the end of incubation (**B**) show that only mutants PL2D4, PL4B5, and PL13H10 have significantly lower growth and potentially altered fitness.
(TIF)

**S3 Fig. Immunogenicity of attenuated *Pa* mutants.** RT-qPCR showing *Dpt* expression 20 h post infection with attenuated *Pa* mutants. Asterisks indicate significance relative to *Pa* WT.
(TIF)

**S4 Fig. Within host growth of representative *Pa* LPS mutants. (A-C)** Proliferation of wild-type *Pa* (**A**), PL10C8 (**B**), and PL11H9 (**C**) mutants in wild-type and *Relish* mutant flies. Wild-type *Pa* proliferated monotonically in both wild-type and *Relish* mutant flies (A). The loads of PL10C8 and PL11H9 mutants remained stable in wild-type flies but increased in *Relish* mutant. Results are shown as mean ± SD of 5 samples (individual flies depicted with dots).
(TIF)

**S5 Fig. *Pa* attenuated mutants are virulent to flies lacking AMPs. (A-W)** Survival rates of 23 attenuated *Pa* mutants in WT and *ΔAMP* flies. Vast majority of attenuated mutants retain their virulence in *ΔAMP* flies. See S3 Table for statistical comparisons.
(TIF)

**S6 Fig. *Drosocin* overexpression improves survival of *Relish* flies after infection with *Pa* LPS mutants. (A-O)** Survival rates of *Relish* control line and *Relish* line overexpressing *Drosocin* in fat body after infection with 15 *Pa* LPS mutants. Only mutants PL1C4 (A) and PL13C10 (O) were not affected by *Drosocin* overexpression. The other mutants cause significantly lower mortality in overexpression line. (**P**) Survival rates of *Relish* control line and *Relish* line overexpressing *Drosomycin* in fat body after infection with PL2D4 mutant. No significant improvement of survival was observed in contrast to *Drosocin* overexpression in panel B.
(TIF)

**S7 Fig. Disk diffusion assay with H2O2.** Representative pictures of agar plates showing the size of inhibition zones for WT *Pa*, ROS-sensitive mutant PL1A3, ROS-resistant mutant

PL5D5.
(TIF)

**S8 Fig. Hemocytes but not melanisation contribute to the defense against *Pa* infection.** (**A-D**) ROS sensitive *Pa* mutants remain less virulent to melanisation-deficient *PPO1*,*2* flies, suggesting that melanisation has no major role in the defense against *Pa* and is unlikely source of ROS. (**E-I**) Genetic ablation of hemocytes significantly increases susceptibility to infection with ROS-sensitive *Pa* mutants.
(TIF)

**S1 Table. Lists of differentially-expressed genes detected by RNA-seq 6h and 20h post *Pa* infection.**
(XLSX)

**S2 Table. Minimal inhibitory concentrations for AMPs and antibiotics determined using WT *Pa*.**
(XLSX)

**S3 Table. Summary statistics for survivals shown in Figs 4 and S3.**
(XLSX)

## Acknowledgments

We are grateful to Bruno Lemaitre and the Bloomington Drosophila Stock Center (NIH P40OD018537) for fly stocks. We thank Francesca Müller-Marquardt for technical help at early stages of transposon mutant library creation and LPS-BIOSCIENCES for help with LPS analysis. We thank the Max Planck-Genome-centre Cologne (http://mpgc.mpipz.mpg.de/home/) for performing RNA-seq in this study.

## Author Contributions

**Conceptualization:** Igor Iatsenko.

**Data curation:** Maria Shaka, Aranzazu Arias-Rojas, Alexandra Hrdina, Igor Iatsenko.

**Formal analysis:** Aranzazu Arias-Rojas, Alexandra Hrdina, Igor Iatsenko.

**Funding acquisition:** Igor Iatsenko.

**Investigation:** Maria Shaka, Aranzazu Arias-Rojas, Alexandra Hrdina, Dagmar Frahm.

**Methodology:** Aranzazu Arias-Rojas.

**Project administration:** Igor Iatsenko.

**Resources:** Maria Shaka.

**Supervision:** Igor Iatsenko.

**Validation:** Maria Shaka.

**Visualization:** Aranzazu Arias-Rojas, Alexandra Hrdina, Igor Iatsenko.

**Writing – original draft:** Igor Iatsenko.

**Writing – review & editing:** Aranzazu Arias-Rojas, Alexandra Hrdina, Igor Iatsenko.

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
