## [Decision Letter · Decision Letter 0]

30 Jun 2022

Dear Dr Iatsenko,

Thank you very much for submitting your manuscript "Lipopolysaccharide-mediated resistance to host antimicrobial peptides and hemocyte-derived reactive-oxygen species are the major Providencia alcalifaciens virulence factors in Drosophila melanogaster" for consideration at PLOS Pathogens. As with all papers reviewed by the journal, your manuscript was reviewed by members of the editorial board and by several independent reviewers. In light of the reviews (below this email), we would like to invite the resubmission of a significantly-revised version that takes into account the reviewers' comments.

Your work has been reviewed by experts in both the field of Drosopphila and bacteria. They recognized the quality of the work and in particular the study of the response of the model host used in this study. Two of the experts ask for a more detailed analysis of the bacterial mutants used and if possible a more directed mutagenesis approach with the possibility to complement these mutants. One of the reviewers also asked that the originality of this work be put into perspective in the discussion, knowing that the role of LPS has been widely studied in the virulence of Gram-negative pathogenic bacteria. Other comments are more suggestions were provided to improve the manuscript.

We cannot make any decision about publication until we have seen the revised manuscript and your response to the reviewers' comments. Your revised manuscript is also likely to be sent to reviewers for further evaluation.

Sincerely,

Eric Oswald, Ph.D., D.V.M.

Associate Editor

PLOS Pathogens

Brian Coombes

Section Editor

PLOS Pathogens

Kasturi Haldar

Editor-in-Chief

PLOS Pathogens

orcid.org/0000-0001-5065-158X

Michael Malim

Editor-in-Chief

PLOS Pathogens

orcid.org/0000-0002-7699-2064

Your work has been reviewed by experts in both the field of Drosopphila and bacteria. If they recognize the quality of the work and in particular the study of the response of the model host used in this study, two of the experts ask for a more detailed analysis of the bacterial mutants used and if possible a more directed mutagenesis approach with the possibility to complement these mutants. One of the reviewers also asked that the originality of this work be put into perspective in the discussion, knowing that the role of LPS has been widely studied in the virulence of Gram- pathogenic bacteria. Other comments are more suggestions to improve the manuscript.

Reviewer's Responses to Questions

**Part I - Summary**

Reviewer #1: In their publication, Shaka and collaborators showed by “forward” and “reverse” approaches that LPS-mediated resistance to host antimicrobial peptides and hemocyte-derived reactive-oxygen species are the major Providencia alcalifaciens virulence factors in Drosophila melanogaster. The paper is very-well written and the hypothesis are well-driven. However, I have major points that are concerning for this paper to be published in PLoS Pathogens.

Reviewer #2: In this manuscript, Shaka and colleagues describe virulence mechanisms of Providencia alcalifaciens in a Drosophila melanogaster infection model. P. alcalifaciens is an opportunistic pathogen of humans and other Providencia species are frequently used in the study of Drosophila immunity. The authors use genetic manipulation of both bacteria and host to demonstrate that P. alcalifaciens uses LPS modification to evade AMP-mediated host defenses and ROS detoxification as an anti-immune mechanism, arguing that these mechanisms of negating host defenses constitute major virulence mechanisms of P. alcalifaciens in D. melanogaster. The authors also find that host sequestration of iron also plays a major role in controlling P. alcalifaciens infection, and I wonder why they do not emphasize that more in the in Abstract. The work presented is thorough and makes elegant use of reciprocal genetic manipulation in host and pathogen. The findings are novel and interesting, and the article is very well written.

Reviewer #3: This MS by Shaka et al. exploits the power of Drosophila as a model to study host-pathogen interactions. The authors study Providencia alcalifaciens virulence and the host immune responses essential to control this pathogen. The strengths of the work are the unbiased and exhaustive functional approaches deployed by the authors and the robustness, clarity and span of the phenotypes observed. I only have very minor comments listed below. This work is of the highest standard in the host-pathogen field and illustrate another time the power of the fly model to probe in unbiased ways the key mechanisms shaping host-pathogen interactions outcomes both in the host and in the pathogen.

**Part II – Major Issues: Key Experiments Required for Acceptance**

Reviewer #1: The main message is the demonstration that LPS and ROS resistance are crucial for virulence of Providencia alcalifaciens. However, LPS and ROS resistance are always associated with the pathogenicity of Gram negative pathogens and this has been highlighted by the paper of Yuan et al., (https://doi.org/10.3389/fmicb.2020.572642) in their comparative genomic analysis. To go further and publish in PLoS Pathogens, this story needs to be decipher with at least the demonstration with isogenic mutants of Pa corresponding to their transposon and the respective complementation. It was already known that Polymyxin B targets LPS (e.g. https://doi.org/10.1039/D1FD00036E). Then I don’t get the point of the authors to use Polymyxin B as a AMP to show that LPS is linked to AMP resistance? Does Drosophila produce an AMP that shows a similar mode of action than Polymyxin B?

Reviewer #2: The authors use RNAseq to document activation of host immune responses to infection. The RNAseq data are not described in much detail and therefore do not give much insight beyond what was shown with the qPCR data. However, the presented analysis is sufficient for supporting the authors’ main claim that the host immune system is activated by P. alcalifaciens infection, and deposit of the RNAseq data into public databases will allow subsequent researchers to make more thorough use of them.

The authors use synthetic or purified AMPs to test whether the Pa mutants they recover from their screen are more susceptible to AMPs but find that the AMPs do not restrict bacterial growth in in vitro assays. This was disappointing, I am sure, and I can think of a few possible reasons for it. One is that the tested peptides may indeed have no effect on this bacterium, following the logic of papers like that of Hanson et al (cited ref 57) showing that AMP effects on particular bacteria can be quite specific (indeed, the authors eventually suggest that Drosocin is the primary AMP controlling Pa infection, and Drosocin is not tested in the in vitro assays). Additionally, other authors have shown that in vitro effects of AMPs can be different than in vivo effects of mutants or knockdowns for those same AMPs (see papers from Jens Rolff’s group), suggesting that physiological context or interaction among peptides is important. Furthermore, several investigators from multiple teams informally discuss having had quite a bit of trouble getting in vitro AMP assays to work effectively for a variety of technical reasons (AMPs adhering to plastic assay wells instead of distributing in solution, effects of pH and other chemistry, etc) but this is unfortunately not citable because it is largely unpublished. The authors of the present study do find that the LPS-mutant bacteria are susceptible to polymyxin B, although this AMP is not produced by Drosophila. They also find that Relish mutant hosts are highly susceptible to infection by the mutant Pa, which implies that they are rescued from AMP control in the mutants, but of course Relish also regulates other defense genes in addition to AMPs. The cleanest test of the hypothesis that the LPS-defective mutants are susceptible to AMPs comes from the observation that the Hanson strain of D. melanogaster lacking 10 AMPs and the strain lacking Drosocin specifically are more susceptible to infection by the mutant bacteria. Since these are the definitive experiments, my suggestion would be to move that experiment to the top of the paragraph that begins on Line 298. The authors could subsequently describe the other experiments, including a direct comparison of death rates between Relish and AMP-deletion mutants to demonstrate the AMPs are the major Relish-regulated mechanism of defense. The manuscript could also include some discussion of the limitations of the in vitro assays. These comments are all for the authors to consider and incorporate at their discretion – I am not trying to force a change to the content of the manuscript here, only sharing my perspective on these experiments and my understanding of how they work (or fail to work) in other labs.

The authors do an overexpression rescue with Drosocin and conclude that the rescue means that Drosocin is “particularly important against Pa infection”. A valuable control here would be to test overexpression of other AMPs and show that they do not rescue. It is possible that overexpression of any AMP above physiological levels will control Pa infection and that this effect is not specific to Drosocin. On the other hand, finding that the effect is specific to Drosocin would be a very compelling result that would complement the observation of susceptibility of the specific Drosocin deficiency.

The experiments documenting the contributions of ROS to host defense and the susceptibility of the Pa mutants to host ROS are nicely performed and I am convinced by them. I appreciate the multiple approaches, including biochemical interventions, the Duox and SOD mutations, and the use of the flies with ablated hemocytes. On line 393, the authors should explicitly state that “hemoless” refers to those flies or eliminate the word from the manuscript since it is not a standard genetic notation.

The authors do a forward genetic screen in P. alcalifaciens to identify mutants that show reduced infectivity and virulence. As a control, the document that the mutants do now show compromised growth in LB. This is an essential control, especially because mutants with altered LPS may well have altered rates of cell division, but the authors should also note bacterial growth inside of a fly might be quite different that growth in LB (different nutrient availability, etc). The ideal experiment would be to measure rate of proliferation inside of a fly, similarly to what was done in cited ref 59 in the study, particularly because small variations in growth rate could have large effects or host survival time. I understand that it would be difficult to separate growth defects from variation in sensitivity to host immune defense in such an experiment, but perhaps it could be approximated by using Drosophila mutants. I recognize that this would be a large effort and I would not recommend doing it for all mutants recovered from the screen. However, at the authors’ discretion, it may be worth doing so for one or two key bacterial mutants representing the processes that they believe are driving resistance to host defenses.

The forward screen did not reveal any effectors such as bacterial toxins, which the authors found surprising (line 503). However, it seems possible that no such mutants were recovered from the screen because toxins are likely to be redundant so individual mutants may not have much phenotype. Nevertheless, one might have expected mutations in secretion systems to have emerged from the screen, particularly since P. alcalifaciens should have an intact Type III Secretion System. The absence of T3SS mutants in the screen could imply that translocation of effectors is not a mechanism that Pa uses to infect Drosophila.

There is no change for the authors to make in the present study, but the use of the w[1118] strain of Drosophila melanogaster as a nominal wildtype is disappointing. They have done so because many of the host mutants they use in this study are generated in the w[1118] background, so from a genetic perspective, the experiments are done correctly. However, the white mutation is known to be massively pleiotropic and the w[1118] strain is susceptible to infection relative to other Drosophila genotypes. Especially because the mutations studied in this work on not on the X chromosome, the better way to have done this experiment would have been to cross the white-mutant X out of the stocks and conduct the study in a wildtype background. I emphasize that I am not asking the authors to do this now or to repeat this study, and this comment is in no way intended to be a block on publication of the current work. I am only asking the authors to consider avoiding this weak genotype in future studies.

Reviewer #3: NA

**Part III – Minor Issues: Editorial and Data Presentation Modifications**

Reviewer #1: 1- line 168 “at OD10 they were all killed within 20h” – This OD should appear in the mat and met

2- line 214 to 217 “Our transcriptomic analysis shows that Drosophila…implying that the high virulence of Pa is not due to the inability of the host to recognize or respond to infection”. Did the author compare with a less “highly virulent” strain?

3- Figure 5: O-antigen. Which band is it?

4- line 354 “prefed”: nothing is written about that in the mat and met. How do the authors perform this experiment?

5- line 558-559 “surviving flies were counted at regular intervals for 5 days after infection”: the authors should specify more precisely what are “regular intervals”.

Reviewer #2: The introduction and discussion are very nicely written. They are clear, comprehensive, scholarly, and easy to read.

Line 131 states that no strains of other Providencia species are intracellularly invasive. This should probably say “no currently described strains” since it is possible that some unstudied strains possess this ability.

Line 173, replace “fast” with “quickly”

Although I understand the argument the authors are making, there is some logical inconsistency in the paragraph that begins on line 219. They show that infected flies have activated Toll and JAK-Stat pathways, but that mutants of these pathways are not more susceptible to infection. Thus, the infection activates the pathway but activation does not contribute to defense. However, Imd pathway mutants are more susceptible to infection so the authors conclude “this is consistent with the upregulation of imd-responsive genes by Pa”. It is internally inconsistent to conclude that the activation does not necessarily imply effect with Toll and JAK-Stat, but that activation is consistent with an expectation of effect in Imd. The more natural conclusion from these results is that transcriptional activation is not, by itself, a good predictor of effect. This would be a valuable point to emphasize for the entire community: the genes that are transcriptionally responsive to a treatment are not necessarily the genes that are driving the physiological response to the treatment. An experimental manipulation is required to demonstrate effect, inference from transcript levels is not sufficient.

The references will need to be formatted prior to publication, including correction of a number of the URLs in the reference list.

Reviewer #3: Minor comments:

-The authors start by using males and then mention females at some point, I guess clarifying from the beginning if the traits under study are sex-biased or not is important. Then the authors may choose to focus on only one sex.

-In Fig3 it would have been nice to test spz mutants for survival assays and I am wondering if the pathogen load is impacted in Toll or Jak/stat pathway mutants.

-Also in some key fly loss of function conditions where survival to Pa is impacted (Fig6D-E, Fig8G) maybe the authors shall use diluted concentration of the pathogens or its mutants in order to better see the phenotypical windows.

-On a more general and semantic note I am wondering is mechanisms allowing better tolerance/resistance of the pathogen to host AMP and ROS activities can be qualified as bona fide bacterial virulence factors…? These mechanisms clearly do contribute to the virulence of the pathogen but I feel they may not be qualified as « virulence factors » per se. Maybe the other shall expend on this semantic issue in their discussion.

PLOS authors have the option to publish the peer review history of their article (what does this mean?). If published, this will include your full peer review and any attached files.

Reviewer #1: No

Reviewer #2: **Yes: **Brian Lazzaro

Reviewer #3: **Yes: **François Leulier
---

## [Editor Report · Decision Letter 1]

23 Aug 2022

Dear Dr Iatsenko,

We are pleased to inform you that your manuscript 'Lipopolysaccharide-mediated resistance to host antimicrobial peptides and hemocyte-derived reactive-oxygen species are the major Providencia alcalifaciens virulence factors in Drosophila melanogaster' has been provisionally accepted for publication in PLOS Pathogens.

Best regards,

Eric Oswald, Ph.D., D.V.M.

Associate Editor

PLOS Pathogens

Brian Coombes

Section Editor

PLOS Pathogens

Kasturi Haldar

Editor-in-Chief

PLOS Pathogens

orcid.org/0000-0001-5065-158X

Michael Malim

Editor-in-Chief

PLOS Pathogens

orcid.org/0000-0002-7699-2064
---

## [Editor Report · Acceptance letter]

3 Sep 2022

Dear Dr Iatsenko,

We are delighted to inform you that your manuscript, "Lipopolysaccharide-mediated resistance to host antimicrobial peptides and hemocyte-derived reactive-oxygen species are the major Providencia alcalifaciens virulence factors in Drosophila melanogaster," has been formally accepted for publication in PLOS Pathogens.

Best regards,

Kasturi Haldar

Editor-in-Chief

PLOS Pathogens

orcid.org/0000-0001-5065-158X

Michael Malim

Editor-in-Chief

PLOS Pathogens

orcid.org/0000-0002-7699-2064